# Learning distributed representations with efficient SoftMax normalization

**Lorenzo Dall'Amico**                                    *lorenzo.dallamico@isi.it*
*ISI Foundation*

**Enrico Maria Belliardo**                                *enrico.belliardo@isi.it*
*ISI Foundation*

**Reviewed on OpenReview:** *https://openreview.net/forum?id=9M4NKMZOPu*

## Abstract

Learning distributed representations, or embeddings, that encode the relational similarity patterns among objects is a relevant task in machine learning. A popular method to learn the embedding matrices $X, Y$ is optimizing a loss function of the term $\text{SoftMax}(XY^T)$. The complexity required to calculate this term, however, runs quadratically with the problem size, making it a computationally heavy solution. In this article, we propose a linear-time heuristic approximation to compute the normalization constants of $\text{SoftMax}(XY^T)$ for embedding vectors with bounded norms. We show on some pre-trained embedding datasets that the proposed estimation method achieves higher or comparable accuracy with competing methods. From this result, we design an efficient and task-agnostic algorithm that learns the embeddings by optimizing the cross entropy between the softmax and a set of probability distributions given as inputs. The proposed algorithm is interpretable and easily adapted to arbitrary embedding problems. We consider a few use cases and observe similar or higher performances and a lower computational time than similar "2Vec" algorithms.

## 1 Introduction

A foundational role of machine learning is providing complex objects with expressive vector representations that preserve some properties of the represented data. These vectors, also known as *embeddings* or *distributed representations*, enable otherwise ill-defined operations on the data, such as assessing their similarity (Bengio et al., 2013; Bellet et al., 2015). Relational patterns among the represented objects are a relevant example of a property that embedding vectors can efficiently encode. For instance, in language processing, distributed representations capture the semantic similarity between words, leveraging the patterns in which they appear in a text. One way of achieving this result is by training the softmax scores $\text{SoftMax}(XY^T)$, where $X, Y$ are two embedding matrices, possibly with $X = Y$. The softmax scores range between 0 and 1 and can be used to promote similar representations for related objects. The optimization of the softmax scores is at the core of all "`2Vec`"-type algorithms but is also key in attention-based transformers (Vaswani et al., 2017).

A known limitation of optimizing the softmax scores is the computational complexity, which scales quadratically with the system size and is impractical for large datasets. As a turnaround, Mikolov et al. (2013) considered two approximation strategies to train the embedding vectors, namely hierarchical softmax (Goodman, 2001; Morin & Bengio, 2005), and *negative sampling*. Thanks to its efficiency, negative sampling is largely adopted and several works proposed analyses or extensions (Mu et al., 2019; Rawat et al., 2019; Shan et al., 2018; Bamler & Mandt, 2020), but also highlighted the weaknesses of this approach (Landgraf & Bellay, 2017; Chen et al., 2018; Qin et al., 2016; Mimno & Thompson, 2017). Both efficient implementations of `2Vec`-type algorithms, however, consider *de facto* modified loss functions with respect to the originally proposed one, thus circumventing the computational bottleneck. A large body of literature, instead, focused on and is still actively exploring methods to approximate and train softmax efficiently. These works include

importance sampling methods (Bengio & Senécal, 2003; Blanc & Rendle, 2018; Rawat et al., 2019; Baharav et al., 2024), random feature methods based on the kernel trick (Rahimi & Recht, 2007; Choromanski et al., 2020; Peng et al., 2021), low-rank approximations (Drineas et al., 2005; Shim et al., 2017; Xiong et al., 2021), and local hashing techniques (Mussmann et al., 2017; Zaheer et al., 2020; Beltagy et al., 2020).

In this work, we provide a closed formula to approximate the softmax normalization constants in linear time, with theoretical and empirical results supporting our method. We evaluate the accuracy of the proposed approximation on pre-trained embeddings and compare it with several related methods. Building on this result, we design an efficient embedding algorithm in the `2Vec` spirit, training a loss function containing the term $\log[\mathrm{SoftMax}(XY)]$. Given their good scalability and simplicity of design, the `2Vec` algorithms proved extremely useful for machine learning users and have been applied to various contexts. However, we recall that `2Vec` algorithms do not optimize the proposed loss function in their most popular implementation. Our algorithm, instead, optimizes the original loss function efficiently and leverages the existing theoretical works that analyzed it, like (Jaffe et al., 2020). We note that our proposed approximation of the softmax score normalization is sufficient to efficiently compute $\log[\mathrm{SoftMax}(XY^T)]$ because this term is composed of two parts: the numerator $XY^T$ that is low rank and can consequently be efficiently optimized; the denominator involving the softmax normalization constants that is the computational bottleneck. Our estimation formula allows us to compute it in $\mathcal{O}(n)$ operations, and thus to efficiently optimize the embedding cost function. We showcase a few practical applications evidencing that our algorithm is competitive or outperforms comparable methods in speed and performance.[1] We recall that all these benchmark algorithms have a linear complexity in $n$, and do not optimize the softmax function as we do. We stress that we do not claim sampling approaches are bad *per se*, but rather, we investigate how to optimize a cost function containing the softmax scores efficiently. A `Python` implementation of our algorithm is available at github.com/lorenzodallamico/EDRep.

## 2 Main result

This section describes how to efficiently approximate the normalization constants of $\mathrm{SoftMax}(XY^T)$, where $X \in \mathbb{R}^{n \times d}$, $Y \in \mathbb{R}^{m \times d}$ are two embedding matrices, with a potentially different number of rows, but with the same number of columns. We denote with $\{\boldsymbol{x}_i\}_{i=1,\dots,n}$ and $\{\boldsymbol{y}_a\}_{a=1,\dots,m}$ the sets of vectors contained in the rows of $X$ and $Y$, respectively. The $i$-th softmax score normalization constant reads

$$Z_i = \sum_{a=1}^{m} e^{\boldsymbol{x}_i^T \boldsymbol{y}_a} \ . \tag{1}$$

In the remainder, we let $m = \mathcal{O}(n)$, implying that computing all the $Z_i$'s requires $\mathcal{O}(dnm) = \mathcal{O}(dn^2)$ operations, which is prohibitively expensive for large datasets. This condition determines the relevance of our problem setting; if $m \ll n$, one can efficiently compute the normalization constants. We further assume $d \ll n$, a necessary condition to estimate all $Z_i$ in linear time. We remark that other works, such as Baharav et al. (2024), have considered a complementary view to ours, by estimating the softmax normalization in sub-linear time in $d$. This setting is relevant for high-dimensional embedding vectors with $d = O_n(n)$.

### 2.1 Linear-time softmax normalization

We consider a vector $\boldsymbol{x}_i \in \mathbb{R}^d$ fixed (corresponding to the $i$-th row of $X$) and treat $\{\boldsymbol{y}_1, \dots, \boldsymbol{y}_m\}$ as random independent vectors drawn from an unknown distribution. This allows us to study $Z_i$ as a random variable and describe its concentration properties. Note that we are not assuming the embedding dimensions to be independent, thus preventing the representation of complex similarity patterns. Such relations are captured by the underlying unknown probability distribution on which we make no assumption besides requiring embedding vectors to have bounded norms. We derive our approximation in three steps. In the first, we show the concentration of $Z_i/m$ around its expectation for which we obtain an integral form, depending on the unknown distribution $f_i$ of the scalar product $\boldsymbol{x}_i^T \boldsymbol{y}_a$. In the second, we introduce a variational approximation by substituting the unknown distribution of the scalar product $f_i$ with a Gaussian distribution. In the third, we introduce a clustering-based method applied to the rows of $Y$ to improve the estimation accuracy.

---

[1]All codes are run on a Dell Inspiron laptop with 16 GB of RAM and with a processor 11th Gen Intel Core i7-11390H @ 3.40GHz × 8.

**Concentration properties of the normalization constants**

We succinctly describe the main concentration result of the normalization constants $Z_i$, which we formally enunciate and prove in Appendix A. We denote with $f_{i,m}$ the unknown empirical distribution of the scalar product $\boldsymbol{x}_i^T \boldsymbol{y}_a$ and suppose its support is in $[-h, h]$ for $h = O_m(1)$. Assuming the convergence in distribution of $f_{i,m}$ to $f_i$, with high probability

$$\lim_{m \to \infty} \frac{Z_i}{m} = \int_{-h}^{h} dt \ e^t f_i(t) \ . \tag{2}$$

Our result holds unchanged also in the case $X = Y$. We remark that we do not require any assumptions on the embedding vectors' distribution (besides having finite norms), nor that the embedding vectors are identically distributed. For a more accurate enunciation, we refer the reader to Theorem A.1. The constant $h$ controls the concentration speed of $Z_i/m$ around its mean: a smaller $h$ implies a faster convergence. The assumption $h = \mathcal{O}_m(1)$ guarantees good concentration properties, and it can be easily enforced by considering embedding vectors with bounded norms. Equation 2 is obtained by combining Theorem A.1 with Theorem A.3 in which we show that the $\mathbb{E}[Z_i]/m$ converges to the integral form on the right hand-side. This is generally not true under the loose assumption of convergence in distribution we formulated.

**Gaussian approximation of the scalar product distribution**

Equation (2) gives a tractable expression of the random variable $Z_i$, but it cannot be solved without additional hypotheses. We thus proceed by introducing a variational approximation of $f_i$, denoted with $\tilde{f}_i$. Thanks to Theorem A.3, the goodness of this approximation only needs to hold in distribution sense, i.e., the cumulative density functions of $f_i$ and $\tilde{f}_i$ should be close. We choose to write $\tilde{f}_i$ as a Gaussian distribution. Recalling that $f_i$ is the distribution of a scalar product – i.e., a sum of random variables – we expect that for $d$ large enough, it is well approximated by the normal distribution. As shown in Section 2.2, the empirical evidence confirms the goodness of this approximation on real data. We denote with $\boldsymbol{\mu}, \Omega$ the mean and covariance of $\boldsymbol{y}$ and with $\mathbb{E}[\cdot]$ the expectation over $\boldsymbol{y}$. We obtain $\mathbb{E}[\boldsymbol{x}_i^T \boldsymbol{y}] = \boldsymbol{x}_i^T \boldsymbol{\mu}$ and $\mathbb{V}[\boldsymbol{x}_i^T \boldsymbol{y}] = \boldsymbol{x}_i^T \Omega \boldsymbol{x}_i$ and write

$$\lim_{m \to \infty} \frac{Z_i}{m} = \int_{-h}^{h} dt \ e^t f_i(t) \approx \int_{\mathbb{R}} dt \ e^t \mathcal{N}(t; \boldsymbol{x}_i^T \boldsymbol{\mu}, \boldsymbol{x}_i^T \Omega \boldsymbol{x}_i) = \exp\left\{ \boldsymbol{x}_i^T \boldsymbol{\mu} + \frac{1}{2} \boldsymbol{x}_i^T \Omega \boldsymbol{x}_i \right\} \ . \tag{3}$$

Note that this approximation does not require the embedding vectors to be drawn from a multivariate Gaussian distribution. As a remark, we move from an integral on $[-h, h]$ to one over the real axis, but the contribution from the tails is negligible since the integrand goes to zero at least as fast as $e^{-|t|}$. The considerable advantage of Equation (4) is that, given $\boldsymbol{\mu}$ and $\Omega$, the normalization constant $Z_i$ is computed in $\mathcal{O}(d^2)$ operations, independently of $m$. This allows us to estimate all $Z_i$'s in $\mathcal{O}(n)$ operations. Since $\boldsymbol{\mu}$ and $\Omega$ are both estimated in $\mathcal{O}(n)$ operations, this formula allows the linear-time estimation of the normalization constants. We note the quadratic complexity in $d$, but, since we are working under the assumption $d \ll n$, this does not hamper the computational efficiency of the proposed method.

**Multivariate Gaussian approximation of the scalar product distribution**

To obtain Equation (3), we assumed all embedding vectors to be well described by the same distribution. In practice, it is common that the represented objects form clusters in the embedded space that mirror affinity groups. We can leverage these clusters to improve the estimation accuracy by clustering the embedding vectors to reduce the within-group embedding variance. We subdivide the set $\mathcal{V}$ of all elements in $\kappa$ non-overlapping subsets $\mathcal{V}_{\alpha=1,\ldots,\kappa}$. The normalization constant then reads

$$Z_i = \sum_{a=1}^{m} e^{\boldsymbol{x}_i^T \boldsymbol{y}_a} = \sum_{\alpha=1}^{\kappa} \underbrace{\sum_{a \in \mathcal{V}_\alpha} e^{\boldsymbol{x}_i^T \boldsymbol{y}_a}}_{\bar{Z}_{i\alpha}} = \sum_{\alpha=1}^{\kappa} \bar{Z}_{i\alpha} \ .$$

If $\mathcal{V}_\alpha = O_m(m)$ for all $\alpha$, then $\bar{Z}_{i\alpha}$ respects the same concentration properties of Equation (2). This assumption is also necessary to keep a low computational complexity. We update Equation (3) as follows

$$\frac{Z_i}{m} \approx \sum_{\alpha=1}^{\kappa} \pi_\alpha \ e^{\boldsymbol{x}_i^T \boldsymbol{\mu_\alpha} + \frac{1}{2} \boldsymbol{x}_i^T \Omega_\alpha \boldsymbol{x}_i} \ , \tag{4}$$

where $\boldsymbol{\mu}_\alpha, \Omega_\alpha$ are the mean vector and covariance matrix of the vectors $\boldsymbol{y}$ in class $\alpha$, and $\pi_\alpha = |\mathcal{V}_\alpha|/m$.

$$\pi_\alpha = \frac{|\mathcal{V}_\alpha|}{m}; \quad \boldsymbol{\mu}_\alpha = \frac{1}{|\mathcal{V}_\alpha|} \sum_{i \in \mathcal{V}_\alpha} \boldsymbol{x}_i; \quad \Omega_\alpha = \frac{1}{|\mathcal{V}_\alpha| - 1} \sum_{i \in \mathcal{V}_\alpha} (\boldsymbol{x}_i - \boldsymbol{\mu}_\alpha)(\boldsymbol{x}_i - \boldsymbol{\mu}_\alpha)^T \ . \tag{5}$$

We choose the partition of $\mathcal{V}$ to minimize the variance $\boldsymbol{x}_i^T \Omega_\alpha \boldsymbol{x}_i$ within each class. We define the total variance $V$ and show the following relation with the *k-means* objective (MacQueen, 1967).

$$V = \sum_{\alpha=1}^{\kappa} \sum_{a \in \mathcal{V}_\alpha} \boldsymbol{x}_i^T \Omega_\alpha \boldsymbol{x}_i \rightarrow \sum_{\alpha=1}^{\kappa} \sum_{a \in \mathcal{V}_\alpha} \left[ \boldsymbol{x}_i^T (\boldsymbol{y}_a - \boldsymbol{\mu}_\alpha) \right]^2 \leq \|\boldsymbol{x}_i\|^2 \sum_{\alpha=1}^{\kappa} \sum_{a \in \mathcal{V}_\alpha} \|\boldsymbol{y}_a - \boldsymbol{\mu}_\alpha\|^2 \ .$$

Using *k-means* to obtain the clusters $\mathcal{V}_{\alpha=1,\dots,\kappa}$, we minimize the upper bound of the total variance, hence improving the estimation performance. The complexity of Lloyd algorithm (Lloyd, 1982) to perform *k-means* clustering scales as $\mathcal{O}(nd\kappa)$. As such, with Equation (4) all $Z_i$s can be computed in $\mathcal{O}(n\kappa d^2)$ operations.

## 2.2 Empirical evaluation

We consider 6 datasets taken from the NLPL word embeddings repository[2] (Kutuzov et al., 2017), representing word embeddings obtained with different algorithms and trained on different corpora:

0. *British National Corpus*; Continuous Skip-Gram, $n = 163.473$, $d = 300$;
7. *English Wikipedia Dump 02/2017*; Global Vectors, $n = 273.930$, $d = 300$;
16. *Gigaword 5th Edition*; fastText Skipgram, $n = 292.967$, $d = 300$;
30. *Ancient Greek CoNLL17 corpus*; Word2Vec Continuous Skip-gram, $n = 45742$, $d = 100$;
187. *Taiga corpus*; fastText Continuous Bag-of-Words, $192.415$, $d = 300$;
224. *Ukrainian CoNLL17 corpus*; Continuous Bag-of-Word, $n = 99.884$, $d = 200$.

The number reported in the list above corresponds to the ID used in the repository. For each dataset, we (1) rescale the embedding vectors so that their average norm equals 1; (2) sample 1000 random indices; (3) compute the corresponding exact and the estimated $Z_i$ values for different approximation orders $\kappa$. We then repeat the same procedure by imposing that all embedding vectors have unitary norms. Figure 1 shows the results of this procedure. The first column compares the empirical distribution cumulative density functions (CDF) of $f_i$ and $\tilde{f}_i$ obtained for $\kappa = 5$. Here, $i$ is a randomly selected node and $X = Y$, and we use the rescaled version of the embedding matrix. The third column shows the same result for the normalized embedding matrices. The plots confirm that in all cases, the multivariate Gaussian approximation we introduced achieves high accuracy in estimating $f_i$. The second and fourth columns of Figure 1 evaluate the accuracy of the estimation method, by showing the cumulative density function of the error between the exact and the estimated values of $Z_i$. As expected, the precision of our method increases with $\kappa$.

### Performance comparison

We compare the accuracy of our estimation method with several competing approaches. For all methods, we set the hyper-parameters to have a comparable execution time with our method.

- **Sampling.** A simple way of estimating the normalization constant $Z_i$ efficiently is sampling $g \ll m$ indices of the sum appearing in Equation (1). Small values of $g$ improve the algorithm's speed but lead

---

[2]The datasets can be found at http://vectors.nlpl.eu/repository/ and are shared under the CC BY 4.0 license.

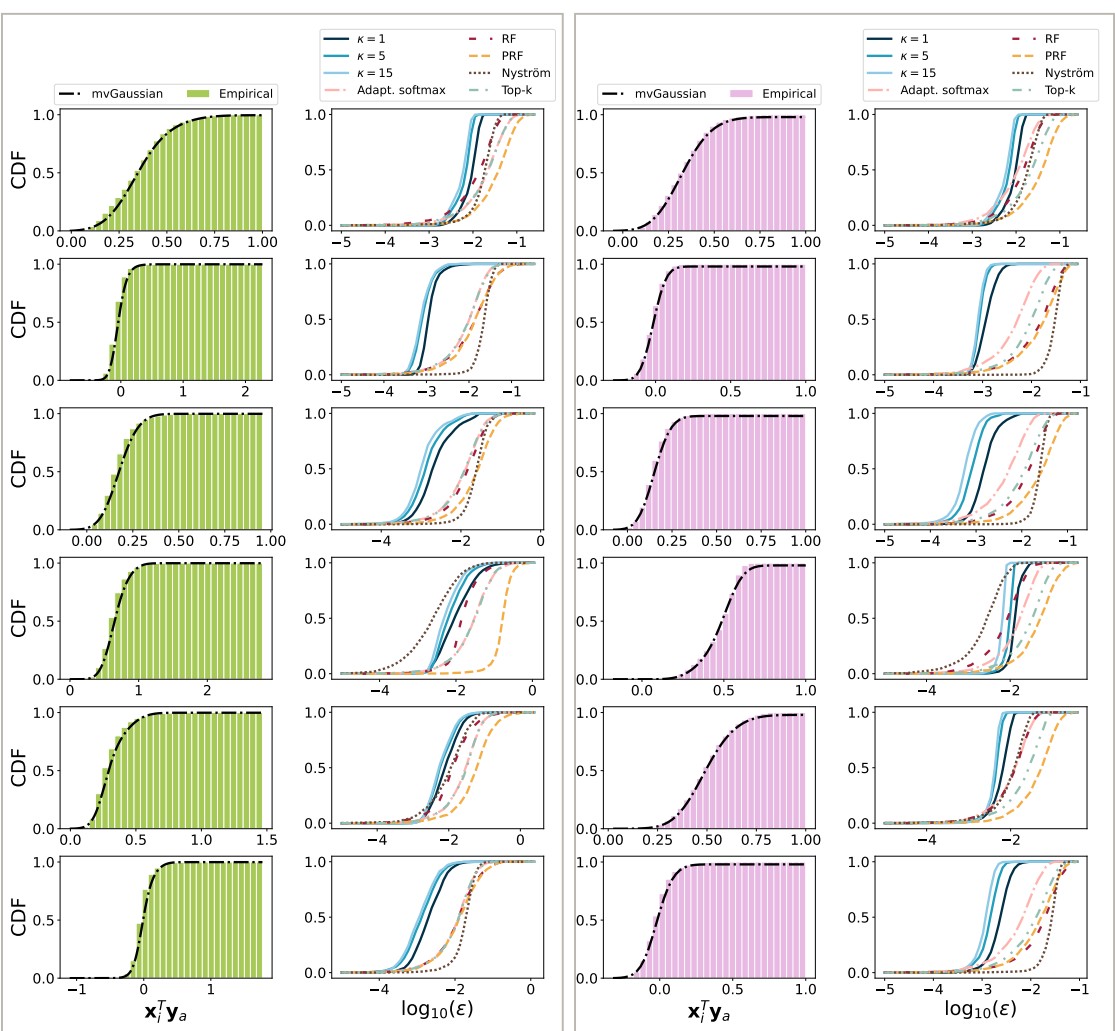

Figure 1: **Evaluation of the softmax normalization constants approximation.** Each row corresponds to one of the six embeddings listed in Section 2.2 which appear in the same order. The first two columns consider the rescaled embedding matrices in which the average norms are equal to one. In the last two, all embedding vectors are normalized to one. The first and third columns show the cumulative density function (CDF) of the scalar product distribution $f_i$ for an arbitrary index $i$. The black dashed-dotted curve is the estimated multivariate Gaussian approximation $\tilde{f}_i$ for $\kappa = 5$. The second and fourth columns show the CDF of the error $\epsilon = |Z - Z_{\text{est}}|/n$ in estimating the softmax score normalization constants. The solid lines refer to our proposed method for three values of $\kappa$ (color-coded). The other lines refer to the methods described in Section 2.2: "Adapt. softmax", pink dashed-dotted line (Blanc & Rendle, 2018); "RF", red loosely-dashed line (Rahimi & Recht, 2007); "PRF", orange dashed line (Choromanski et al., 2020); "Nyström", gray dotted line (Drineas et al., 2005); "Top-k", green dashed double dotted line (Mussmann et al., 2017).

to higher variance estimates. In (Bengio & Senécal, 2003), the authors showed that importance sampling reduces the estimation variance and proposed an unbiased estimator that, however, requires the exact softmax distribution. The authors of (Blanc & Rendle, 2018) proposed *adaptive softmax* in which the sampling probability of index $a$ is proportional to $1 + \alpha(\boldsymbol{x}_i^T \boldsymbol{y}_a)^2$. Leveraging a tree structure and the kernel trick, they further showed how to compute the softmax normalization constants in $\mathcal{O}(d^2 n \log n)$ operations. This algorithm's performance is denoted by "Adapt. softmax" in Figure 1.

- **Random features.** The terms $e^{\boldsymbol{x}_i^T \boldsymbol{y}_a}$ are closely related to the Gaussian kernel $e^{-\|\boldsymbol{x}_i - \boldsymbol{y}_a\|^2}$ and the random feature methods exploit the kernel trick to linearize the softmax score matrix. One defines a random matrix $W \in \mathbb{R}^{D \times d}$ with entries distributed according to $\mathcal{N}(\boldsymbol{0}_d, I_d)$, and a mapping, $\phi_W : \mathbb{R}^d \to \mathbb{R}^{2D}$ so that $\mathbb{E}_W[(\phi_W(\boldsymbol{x}))^T \phi_W(\boldsymbol{y})] = e^{-\frac{\|\boldsymbol{x} - \boldsymbol{y}\|^2}{2}}$. This approximation allows one to compute $Z_i$ efficiently:

$$Z_i = \sum_{a=1}^{m} e^{\boldsymbol{x}_i^T \boldsymbol{y}_a} = e^{\frac{\|\boldsymbol{x}_i\|^2}{2}} \sum_{a=1}^{m} e^{\frac{\|\boldsymbol{y}_a\|^2}{2}} \, \mathbb{E}\left[(\phi_W(\boldsymbol{x}_i))^T \phi_W(\boldsymbol{y}_a)\right] = e^{\frac{\|\boldsymbol{x}_i\|^2}{2}} \, \mathbb{E}\left[(\phi_W(\boldsymbol{x}_i))^T \boldsymbol{m}_W\right] \ ,$$

  where $\boldsymbol{m}_W = \sum_{a=1}^{k} e^{\frac{\|\boldsymbol{y}_a\|^2}{2}} \phi_W(\boldsymbol{y}_a)$ needs to be computed only once. The complexity of these methods runs as $\mathcal{O}(nD)$. In Figure 1, the line denoted as "RF" considers the mapping $\phi$ using random features as proposed in (Rahimi & Recht, 2007), while "PRF" is the line obtained for the *positive random features* proposed in (Choromanski et al., 2020) Both methods are implemented for $D = 1000$.

- **Low-rank approximations.** Letting $Q_{ia} = e^{-\|\boldsymbol{x}_i - \boldsymbol{y}_a\|^2}$, the softmax normalization constant reads $Z_i = (Q \boldsymbol{1}_m)_i$. If $Q$ can be written as the product of two low-rank matrices $A, B$, then $Z_i = AB^T \boldsymbol{1}_m$ can be computed efficiently by multiplying from right to left, without materializing $Q$. One way of performing such a low-rank approximation is to use the Nyström method (Drineas et al., 2005), which we consider for $X = Y$, leading to a symmetric $Q$. We sample $g$ indices and compute the corresponding rows and columns of $Q$. Without loss of generality, we let the sampled indices be the first $g$ and write

$$Q = \begin{pmatrix} Q_{\text{ss}} & Q_{\text{us}}^T \\ Q_{\text{us}} & Q_{\text{uu}} \end{pmatrix} \approx \begin{pmatrix} Q_{\text{ss}} & Q_{\text{us}}^T \\ Q_{\text{us}} & Q_{\text{us}} Q_{\text{ss}}^\dagger Q_{\text{us}}^T \end{pmatrix} \ .$$

  The matrix $Q_{\text{uu}} \in \mathbb{R}^{(n-g) \times (n-g)}$ represents the entries of the unsampled indices and is approximated with $Q_{\text{us}} Q_{\text{ss}}^\dagger Q_{\text{us}}^T$, where $\dagger$ denotes the Moore-Penrose pseudo-inverse. The matrix $Q_{\text{ss}}^\dagger \in \mathbb{R}^{g \times g}$ is small, while $Q_{\text{us}} \in \mathbb{R}^{(n-g) \times g}$. Assuming $g \ll n$, this method estimates all $Z_i$ in $\mathcal{O}(ndg)$ operations. The curve "Nyström" in Figure 1 reports its performance for $g = 50$.

- **Top-$k$ approximation.** The sum in Equation (1) is dominated by the large values of $\boldsymbol{x}_i^T \boldsymbol{y}_a$, because of the exponential function. One can leverage the large literature on *maximum inner product search* to find the $k$ closest embedding vectors to $\boldsymbol{x}_i$ and use them to define an estimator of $Z_i$ with lower variance. For instance, following (Mussmann et al., 2017), we let $\mathcal{S}_i$ be a set with the indices corresponding to the $k$ closest vectors to $\boldsymbol{x}_i$ and $\mathcal{T}_i$ a set of $g$ randomly sampled indices from $\mathcal{V} \setminus \mathcal{S}_i$. The estimator of $Z_i$ reads

$$Z_i \approx \sum_{a \in \mathcal{S}_i} e^{\boldsymbol{x}_i^T \boldsymbol{y}_a} + \frac{n - |\mathcal{S}_i|}{|\mathcal{T}_i|} \sum_{a \in \mathcal{T}_i} e^{\boldsymbol{x}_i^T \boldsymbol{y}_a} \ .$$

  Several methods exist to efficiently estimate the set $\mathcal{S}_i$, and all the $Z_i$s can be computed in $\mathcal{O}(nd(k+g))$ operations. The curve "Top-$k$" in Figure 1 shows the result of this method for $k = g = 25$.

The results show our method systematically outperforms the competing ones, with one exception in which the Nyström-based approach provides better results, even if on other datasets it is outperformed by other methods. Most importantly, we observe high performance for $\kappa = 1$. Here, the clustering step can be omitted and is of particular interest, since the estimator is obtained from simple matrix operations.

## 3 EDRep: an algorithm for efficient distributed representations

Building on the results of Section 2, we describe an efficient algorithm – which we name EDRep – to obtain distributed representations in the 2Vec spirit. These algorithms are still extremely popular among machine learning users thanks to their efficiency, scalability, and flexibility. They build on Word2Vec (Mikolov et al., 2013), an algorithm designed for word embeddings, and were adapted to a variety of domains, including graphs (Perozzi et al., 2014; Grover & Leskovec, 2016; Gao et al., 2019; Rozemberczki et al., 2019; Nickel & Kiela, 2017; Narayanan et al., 2017), time (Kazemi et al., 2019), temporal contact sequences (Goyal et al., 2020; Rahman et al., 2018; Nguyen et al., 2018; Sato et al., 2021; Torricelli et al., 2020), biological entities

(Du et al., 2019; Ng, 2017), tweets (Dhingra et al., 2016) and higher order interactions (Billings et al., 2019) among others. `Word2Vec` trains the embedding in an unsupervised fashion by letting words commonly appearing in the same context have a similar representation. The generalizations of `Word2Vec` to other contexts require the "translation" of the input dataset into a text on which `Word2Vec` is then applied. For instance, in `Node2Vec`, a text is created by performing random walks on a graph.

In our formulation, we take a different perspective and consider a probability matrix $P$ as the input of our problem. This matrix encodes relational patterns between object pairs and is well-suited for datasets describing affinity measures among the embedded objects. We formulate a general embedding problem easily customized to an arbitrary setting. The optimal design of the probability matrix $P$ is problem-dependent and beyond the scope of this article. However, we show in a few applications that simple choices lead to comparable results with competing or better `2Vec` algorithms but with a lower computation time.

### 3.1 Problem formulation

We consider a set $\mathcal{V}$ of $n$ items to be embedded. The relational patterns among the objects in $\mathcal{V}$ are encoded by an affinity probability matrix $P \in \mathbb{R}^{n \times n}$. Our goal is to learn an embedding matrix $X \in \mathbb{R}^d$ that preserves the relational patterns encoded in the matrix $P$. To do so, we adopt a variational approach and minimize the cross entropy between the rows of $P$ and of $\mathrm{SoftMax}(XX^T)$ over the embedding vectors. This allows us to learn the best parametric approximation $X$ to fit the input matrix $P$. More formally, we define the embedding matrix $X$ as the solution to the following optimization problem:

$$X = \underset{Y \in \mathcal{U}_{n \times d}}{\arg\min} \sum_{i \in \mathcal{V}} \left[ \underbrace{-\sum_{j \in \mathcal{V}} P_{ij} \log\Big( \mathrm{SoftMax}(YY^T)_{ij} \Big)}_{\mathrm{cross-entropy}} + \underbrace{\frac{1}{n} \boldsymbol{y}_i^T \sum_{j \in \mathcal{V}} \boldsymbol{y}_j}_{\mathrm{regularization}} \right], \tag{6}$$

where $\mathcal{U}_{n \times d}$ denotes the set of all matrices of size $n \times d$ having in their rows unitary vectors. The loss function includes a regularization term that promotes embedding matrices with a centered mean. We empirically observed that this term improves the embedding quality in practical applications. The computational bottleneck of this optimization problem lies in the calculation of the softmax score normalization constants. We denote with $E$ the number of non-zero entries of $P$, with $\mathbf{1}_n$ the all-ones vector of size $n$, and with $\mathrm{tr}(\cdot)$ the trace operator. Then, the optimization problem of Equation (6) can be reformulated as follows:

$$X = \underset{Y \in \mathcal{U}_{n \times d}}{\arg\min} \left[ -\underbrace{\mathrm{tr}\left( Y^T P Y \right)}_{\mathcal{O}(Ed)} + \underbrace{\sum_{i \in \mathcal{V}} \log(Z_i)}_{\mathcal{O}(dn^2)} + \frac{1}{n} \underbrace{\mathrm{tr}\left( Y^T \mathbf{1}_n \mathbf{1}_n Y^T \right)}_{\mathcal{O}(nd)} \right], \tag{7}$$

where the values below the brackets indicate the computational cost required for each element. The derivation of this expression is reported in Appendix B. In many relevant settings, $P$ is a sparse matrix, thus $E \ll n^2$. The calculation of the softmax score normalization constants, instead, requires $\mathcal{O}(dn^2)$ operations regardless of $E$ and is the computational bottleneck. We can thus adopt the approximation introduced in Equation (4) to efficiently optimize this cost function and to define an efficient embedding algorithm.

**Remark 3.1.** *In Equation* (6) *and in the remainder, we focus on square matrices $P$, in which rows and columns are defined over the same set $\mathcal{V}$. The optimization problem in Equation* (6) *can however be generalized to an asymmetric scenario in which the entries $P_{ia}$ are defined for $i \in \mathcal{V}$ and $a \in \mathcal{W}$, thus invoking the term $\mathrm{SoftMax}(XY^T)$, for $X \neq Y$. We provide `Python` codes to obtain the embedding also in this setting.*

Let us now detail the main steps needed to translate the result of Equation (4) into a practical algorithm to produce efficient distributed representations.

### 3.2 Optimization strategy

We obtain the embedding matrix $X$ by optimizing the problem formulated in Equation (6) with stochastic gradient descent. We substitute the approximated values of $Z_i$ introduced in Equation (4) and we let

---

**Algorithm 1** EDRep

---

**Input:** $P \in \mathbb{R}^{n \times x}$ probability matrix encoding similarities; $d$, embedding dimension; $\boldsymbol{\ell} \in \{1, \ldots, \kappa\}^n$ node label vector; $\eta_0$, learning rate; `n_epochs`, number of training epochs
**Output:** $X^{n \times d}$, embedding matrix
$X \leftarrow$ initialize the embedding matrix with random unitary vectors
$\eta \leftarrow \eta_0$ initial learning rate
$\pi_{\alpha=1,\ldots,\kappa} \leftarrow$ as per Equation (5)
**for** $1 \leq t \leq$ `n_epochs` **do**
   $\boldsymbol{\mu}_{\alpha=1,\ldots,\kappa}$ , $\Omega_{\alpha=1,\ldots,\kappa} \leftarrow$ update the parameters as per Equation (5)
   $\{\boldsymbol{g}_i\}_{i \in \mathcal{V}} \leftarrow$ gradient matrix as in Equation (8)
   **for** $1 \leq i \leq n$ **do**
      $\boldsymbol{g}'_i \leftarrow \boldsymbol{g}_i - (\boldsymbol{g}_i^T \boldsymbol{x}_i)\boldsymbol{x}_i$ remove the parallel component
      $\boldsymbol{g}''_i \leftarrow \boldsymbol{g}'_i / \|\boldsymbol{g}'_i\|$ normalize
      $\boldsymbol{x}_i \leftarrow \sqrt{1 - \eta^2} \; \boldsymbol{x}_i - \eta \boldsymbol{g}''_i$; gradient descent step
   **end for**
   $\eta \leftarrow \eta - \frac{\eta_0}{\texttt{n\_epochs}}$ linear update of the learning rate
**end for**

---

$M \in \mathbb{R}^{\kappa \times d}$ have the $\{\boldsymbol{\mu}_\alpha\}_{\alpha=1,\ldots,\kappa}$ values in its rows. We further define $\mathcal{Z} \in \mathbb{R}^{n \times \kappa}$ as

$$\mathcal{Z}_{i\alpha} = \pi_\alpha \exp\left\{ \boldsymbol{x}_i^T \boldsymbol{\mu}_\alpha + \frac{1}{2} \boldsymbol{x}_i^T \Omega_\alpha \boldsymbol{x}_i \right\} .$$

With this notation $Z_i / n = (\mathcal{Z} \mathbf{1}_\kappa)_i$. The $i \in \mathcal{V}$ and $q \in \{1, \ldots, d\}$ gradient component reads

$$g_{iq} = -\underbrace{[(P + P^T)X]_{iq}}_{\mathcal{O}(Ed)} + \frac{2}{n}\underbrace{[\mathbf{1}_n \mathbf{1}_n^T X]_{iq}}_{\mathcal{O}(nd)} + \underbrace{\frac{1}{(\mathcal{Z}\mathbf{1}_\kappa)_i}\left[ \mathcal{Z}M + \sum_{\alpha=1}^{\kappa} \mathcal{Z}_{i\alpha}(X\Omega_\alpha) \right]_{iq}}_{\mathcal{O}(\kappa n d^2)}, \tag{8}$$

where the values underneath the brackets indicate the computational complexity required to compute each addend of the gradient matrix. The derivation of Equation (8) is reported in Appendix C. To keep the normalization, we first compute $\boldsymbol{g}'$ removing the component parallel to $\boldsymbol{x}_i$, then we normalize it and obtain $\boldsymbol{g}''_i$ to finally update the embedding as follows for $0 \leq \eta \leq 1$: $\boldsymbol{x}_i^{\text{new}} = \sqrt{1 - \eta^2} \; \boldsymbol{x}_i - \eta \boldsymbol{g}''_i$, that implies $\|\boldsymbol{x}_i^{\text{new}}\| = 1$. Note that Algorithm 1 requires the labeling vector $\boldsymbol{\ell}$ as an input but it is generally unknown. A workaround consists in running EDRep for $\kappa = 1$ for which $\boldsymbol{\ell} = \mathbf{1}_n$, then run $\kappa$-class clustering *k-means* and rerun EDRep algorithm for the so-obtained vector $\boldsymbol{\ell}$.

### 3.3 Computational complexity

To determine the complexity of Algorithm 1, let us focus on its computationally heaviest steps:

1. *The calculation of $\boldsymbol{\ell}$ if $\kappa > 1$.* This is obtained in $\mathcal{O}(n\kappa d)$ operations with *k-means* algorithm.

2. *The parameters update as per Equation* (5). This step is performed in $\mathcal{O}(n\kappa d^2)$ operations.

3. *The gradient calculation as per Equation* (8). This is obtained in $\mathcal{O}(Ed + n\kappa d^2)$ operations as indicated by the brackets in Equation (8).

The gradient calculation is thus the most expensive operation. Our approximation reduces the complexity required to compute the "$Z$ part" of the gradient from $\mathcal{O}(dn^2)$ to $\mathcal{O}(\kappa n d^2)$, with $\kappa, d \ll n$. The most expensive term thus requires $\mathcal{O}(Ed)$ operations. This complexity is prohibitive for dense matrices, but in typical settings $P$ is sparse and the product can be performed efficiently. Nonetheless, even for large values of $E$, if $P$ can be written as the product (or sum of products) of sparse matrices, $PX$ can still be

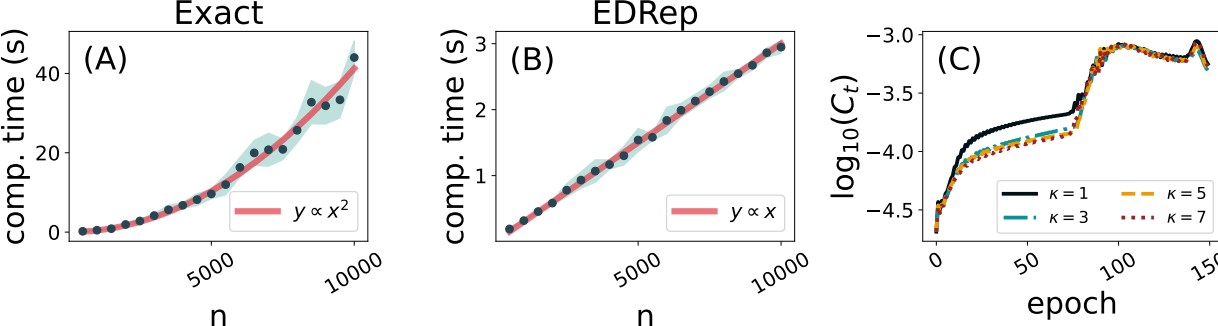

Figure 2: **Comparison with exact gradient calculation**. *Panels A, B*: computation time of the optimization of Equation (6) with gradient descent as function of the size $n$. Panel A refers to the exact gradient, panel B is Algorithm 1. The blue dots are the mean obtained over 10 realizations, the shadow line has the width of one standard deviation. *Panel C*: logarithm of embedding error $C_t = \frac{1}{n}\|X_t X_t^T - \bar{X}_t \bar{X}_t^T\|_F$ between the true and the estimated embedding matrices at training epoch $t$. In this experiment, $n = 3000$. In color code and marker style, we report the results for different values of $\kappa$. The embedding algorithms are run with the same initial condition and parameters: $\eta_0 = 0.7, d = 32, n_{\text{epochs}} = 25$ (for the first two plots).

computed efficiently. In fact, let $P = P_m \cdot P_{m-1} \ldots P_1$ for some positive $m$, then $PX$ can be obtained without materializing $P$, taking the products from right to left:

$$PX = (P_m \cdot P_{m-1} \cdot \overleftarrow{\cdots \cdot P_1 X}) \, ,$$

thus speeding up the computational bottleneck of our algorithm. In our implementation, we explicitly consider this representation of $P$ as an input. When a non-factorized dense matrix $P$ is provided, one could envision adopting a method such as the one presented in (Le Magoarou & Gribonval, 2016) to approximate a dense matrix $P$ with the product of sparse matrices to speed up the algorithm.

### 3.4 Comparison with exact gradient computation

We compare the `EDRep` algorithm described in the previous section with its analogous counterpart in which the gradient of Equation (6) is computed analytically in $\mathcal{O}(n^2 d)$ operations. We considered $P$ as the row-normalized random matrix in which the $(ij)$ entry is set to 1 with probability proportional to $\theta_i \theta_j$ and $\theta_i$ follows a negative binomial distribution with parameters $N = 3$, $p = 0.3$. This choice allows us to generate heterogeneity in the $P$ structure while being capable of controlling the size $n$. Figure 2A shows the computational time corresponding to the exact gradient calculation, while panel B reports the same result for `EDRep`. Let $X_t$ be the `EDRep` embedding at epoch $t$ and $Y_t$ be the corresponding one of the full gradient calculation, we define $C_t = \frac{1}{n}\|X_t X_t^T - Y_t Y_t^T\|_F$, quantifying the deviation between the two embedding methods. Figure 2C shows the behavior of $C_t$ for different $\kappa$ values, evidencing only slight disagreements between the exact and the approximated embeddings that, as expected, decrease with $\kappa$.

## 4 Use cases

We consider a few use cases of our algorithm to test it and showcase its flexibility in practical settings. To perform these tests, we must specify the set $P$ and we adopt simple strategies to define it. We show that, even for our simple choices, the `EDRep` approach achieves competitive (sometimes superior) results in terms of performance with competing `2Vec` algorithms, with a (much) lower computational time.[3] We would like to underline that the sampling probabilities choice is a hard and problem-dependent task and optimally addressing it is beyond the scope of this article. Our aim is not to develop state-of-the-art algorithms for

---

[3]It should be noted that performances typically increase with training time The parameter choice of the `2Vec` algorithms is such that the competing algorithms are comparable on one of the two measures so that the other can be evenly compared.

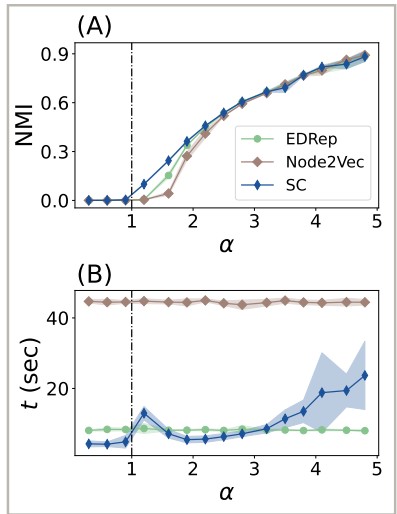 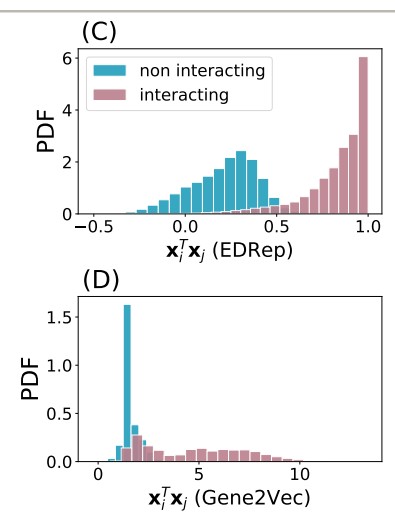 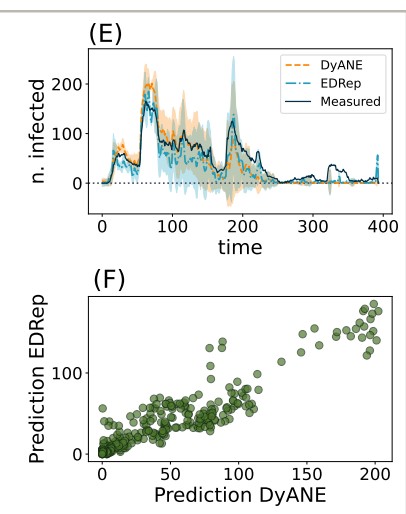

Figure 3: **EDRep use cases**. *First column: community detection. Panel A*: normalized mutual information (NMI) as a function of the problem hardness $\alpha$ (see Appendix D) for a DCSBM graph. We consider graphs with $n = 30\,000$ nodes, expected average degree $c = 10$, and $q = 4$ communities. The green circles refer to the `EDRep` algorithm with $d = 32$, $\kappa = 1$ and $w = 3$, the brown diamonds are `DeepWalk` with $d = 32$, while the blue narrow diamonds are the spectral clustering algorithm of (Dall'Amico et al., 2021). *Panel B*: corresponding computational time in seconds. *Panels A and B*: averages are over 10 samples and the error bar width equals the standard deviation. The two panels share the same legend. *Second column: gene embedding.* Scalar product between gene embedding representation for non-interacting (blue histogram) and interacting (purple histogram) gene pairs. *Panel C*: embedding obtained with `EDRep`; *Panel D* embedding obtained with the `Gene2Vec` algorithm. Panels C and D share the same legend. *Third column: dynamic aware node embedding. Panel E*: number of infected individuals of a SIR process on a proximity network (black solid line) and reconstructed values by `DyANE` (orange dashed line) and using the `EDRep` embedding (blue dashed-dotted line), with $d = 200$ for both methods. The shaded lines are the standard deviations over 50 random trials of the process. *Panel F*: with reference to Panel E, this is the scatter plot between the predicted number of infected per time-stamp by the two strategies.

specific problems but to show that with simple choices we can adapt our algorithm to compete with the closest competing methods in terms of speed and accuracy. For further implementation details regarding the next section, we refer the reader to Appendix D.

## 4.1 Community detection

Graphs are mathematical objects that model complex relations between pairs of items. They are formed by a set of *n nodes* $\mathcal{V}$ and a set of *edges* $\mathcal{E}$ connecting node pairs (Newman, 2003). Graphs can be represented with the *adjacency matrix* $A \in \mathbb{R}^{n \times n}$, so that $A_{ij} = 1$ if $(ij) \in \mathcal{E}$ and equals zero otherwise. A relevant problem in graph learning is *community detection*, the task of determining a non-overlapping node partition, unveiling more densely connected groups of nodes (Fortunato & Hric, 2016). A common way of proceeding – see *e.g.* (Von Luxburg, 2007) – is to create a node embedding encoding the community structure and then clustering the nodes in the embedded space. Following this strategy, we adopt the `EDRep` to produce a node embedding with $P$ being $P = \frac{1}{w} \sum_{t=1}^{w} (L_{\mathrm{rw}})^t$, where $L_{\mathrm{rw}}$ is the *row-normalized adjacency matrix*. The entry $P_{ij}$ is the limiting probability that a random walker on $\mathcal{G}$ goes from node $i$ to $j$ in one $w$ or fewer steps.

We evaluate our algorithm on synthetic graphs generated from the *degree-corrected stochastic block model* (DCSBM) (Karrer & Newman, 2011), capable of creating graphs with a community structure and an arbitrary degree distribution.[4] The inference accuracy is expressed with the normalized mutual information between

---
[4]The degree of a node is its number of connections.

the inferred and the ground truth partition. This score ranges between 0 (random assignment) and 1 (perfect assignment). Figure 3A shows the NMI for different values of $\alpha$, a function of the generative model parameters, controlling for the hardness of the reconstruction problem.[5] The results are compared against other two algorithms that were alternatively deployed to obtain the embedding: the spectral method of (Dall'Amico et al., 2021) that was shown to be nearly Bayes-optimal for this task and `DeepWalk` (Perozzi et al., 2014).[6] The communities are obtained from the embeddings using $k$-class *k-means* clustering.

The results show that the `EDRep`-based algorithm performs almost as well as the optimal algorithm of (Dall'Amico et al., 2021) and a slight mismatch is only observed for $\alpha$ approaching 1. This is a particularly challenging setting in which only a few algorithms can retrieve the community structure. Compared to the `DeepWalk` approach, our method generally yields better results for all $\alpha$. Figure 3B further compares the computation times, giving the `EDRep` approach a decisive advantage with respect to `DeepWalk`. The main advantage with respect to the spectral algorithm, instead, is the algorithm's computational complexity. For a graph with $q$ communities, the considered spectral clustering algorithm runs in $\mathcal{O}(nq^3)$ operations, while the complexity of `EDRep` is independent of $q$.

## 4.2 Gene embeddings

In (Du et al., 2019), the authors develop an algorithm to embed DNA genes from a list of pairs whose co-expression exceeds a threshold value. The dataset comprises 8832 genes and 263016 gene pairs and a list of gene pairs with a binary label indicating whether or not that corresponds to an interacting pair. The `Gene2Vec` algorithm of (Du et al., 2019) builds on `Word2Vec` to obtain meaningful gene vector representations based on their co-expression and uses it to predict pairs of interacting genes.

We obtain the `EDRep` embedding using the row-normalized gene co-occurrence matrix as our choice of $P$. The `Gene2Vec` embedding is generated with the code provided by the authors with default parameters. Both embeddings have dimension $d = 200$. The computation time of `EDRep` is approximately 6 seconds against the 12 seconds needed for `Gene2Vec`. We then train a logistic regression classifier on the embedding cosine similarities with the 70% of the labeled data and test it on the remaining 30% of the data. Our model achieves an accuracy of 92% and outperforms `Gene2Vec` which has an accuracy of 84%. Figures 3(C, D) show the histogram of the cosine similarities between interacting and non-interacting groups that visually explains the performance gap.

## 4.3 Causality aware temporal graph embeddings

In (Sato et al., 2021) the authors describe a method to embed temporal networks while preserving the role of time in defining causality. Temporal networks are represented as a sequence of temporal edges $(i, j, t)$, denoting an interaction between $i$ and $j$ at time $t$. The method relies on the embedding of a *supra-adjacency* matrix, $A_{\text{supra}}$ in $\mathbb{R}^{D \times D}$, where $D = \sum_{t=1}^{T} |\mathcal{V}_t|$ and $\mathcal{V}_t$ is the set of active nodes at time $t$. Here, each node corresponds to a pair "node-time" in the original temporal graph. The *supra-adjacency* matrix is the adjacency matrix of a weighted directed acyclic graph, accounting for time-driven causality. In (Sato et al., 2021) the authors use `DeepWalk` to obtain an embedding from $A_{\text{supra}}$ and use it to reconstruct the states of a partially observed dynamical process taking place on the temporal graph (such as an epidemic spreading) from few observations. Following the same procedure of (Sato et al., 2021), we obtain Figure 3(E-F), in which we compare the reconstruction of an epidemic spreading obtained using `DeepWalk` against `EDRep`, with $P$ being the row-normalization of $A_{\text{supra}}$. The results are barely distinguishable, but `EDRep` is more than 5 times faster than the competing approach.

---

[5]Detection is theoretically feasible if and only if $\alpha > 1$ (Gulikers et al., 2018; 2017).

[6]For the spectral method we used the authors' `Python` implementation available at lorenzodallamico.github.io/codes under the CC BY 4.0 license. For `DeepWalk`, we used the `C++` implementation of github.com/thibaudmartinez/node2vec with its default values, released under the Apache License 2.0.

# 5    Conclusions

This article introduces a linear-time approximation of the softmax scores normalization for embedding vectors with bounded norms. Our result relies on a variational approximation of the unknown scalar product probability distribution between embedding vectors. We provide both theoretical and empirical validation for our estimation formula. By testing our approximation on several empirical datasets, we showed that it can achieve high accuracy levels, outperforming the competing methods.

Based on this result, we describe a simple embedding algorithm in the `2Vec` style with a loss computed in $\mathcal{O}(n^2)$ operations. Because of its complexity, the competing methods use alternative loss functions, while our algorithm efficiently optimizes the original loss function. The proposed `EDRep` algorithm is general-purpose and takes a probability matrix $P$ as input. To prove its efficiency, we tested it on a few use cases and made specific choices for the matrix $P$. The simulations showed that simple and intuitive definitions of such matrix could lead to higher or similar performances compared with competing algorithms. We also observed the `EDRep` algorithm to be systematically faster than its competitors.

Let us now consider some limitations of our work. Given the generality of the formulation, we did not provide a bound to the error on the $Z_i$ estimation. However, we extensively tested our method on several embeddings and matrices $P$, weighted and unweighted, symmetric and not, and with different sparsity levels. In all cases, the results confirmed the goodness of our proposed approach. The reasons justifying these results are two: (1) our approximation needs only to hold in "distribution" sense and we do not need a more stringent point-wise accuracy; (2) the multivariate normal approximation is particularly powerful to approximate the distribution of the scalar product. Unsurprisingly, we also observed that the performance of `EDRep` – compared to the `2Vec` methods – highly depends on the matrix $P$. In some cases, our method provides a neat advantage in terms of performance, while in others the results are essentially identical, with our method being faster. We lack a clear interpretation of the role of $P$ in determining the embedding quality and the convergence speed and we believe this aspect deserves further investigation in the future. We highlight, however, that this analysis is task-dependent, and finding a good $P$ should specifically address a precise research question.

Our approximation is a simple yet efficient method to bypass the quadratic computational complexity required by the softmax normalization. The direct application of this result to the proposed `EDRep` algorithm has several advantages. Despite its simplicity, the practicality of this algorithm makes it particularly appealing to machine learning users who can easily adapt the algorithm to an arbitrary embedding problem. For instance, the `EDRep` algorithm has already been adopted to define the distance between pairs of temporal graphs in (Dall'Amico et al., 2024). Moreover, as observed in (Levy et al., 2015) for word embeddings, tailored fine-tuning, and data preprocessing often have a higher impact in determining the embedding quality than the architecture itself. As such, the `EDRep` constitutes a simple, interpretable, minimal unit to efficiently create distributed representations. Moreover, given the generality of our framework, one can effortlessly adapt `EDRep` to other similar cost functions. The most immediate change is to consider a contrastive learning setting in which also the term $\text{SoftMax}(-XY^T)$ appears. With minor modifications to Algorithm 1, one can also account for non-normalized (but bounded) embedding vectors, or choose $P$ as an arbitrary non-negative matrix. On the other hand, while efficiently dealing with the computation of the softmax scores is crucial in transformer architectures, our results do not directly allow the design of an efficient transformer. However, we envision that ours can be a significant contribution to the design of efficient methods to generate distributed representations.

**Acknowledgments**

LD acknowledges support from the the Lagrange Project of the ISI Foundation funded by CRT Foundation, from the European Union's Horizon 2020 research and innovation programme under grant agreement No. 101016233 (PERISCOPE) and from Fondation Botnar (EPFL COVID-19 Real Time Epidemiology I-DAIR Pathfinder). EMB acknowledges support from CRT Lagrange Fellowships in Data Science for Social Impact of the ISI Foundation. The authors thank Ciro Cattuto for fruitful discussions and Cosme Louart for his valuable feedback on the proofs of Theorems A.1, A.3.

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

## A   Concentration theorems

In this appendix we provide the enunciate and prove the Theorems motivating Equation (2). The first result describes the concentration of $Z_i/m$ around its mean.

**Theorem A.1.** *Consider a vector $\boldsymbol{x}_i \in \mathbb{R}^d$ and a set $\{\boldsymbol{y}_1, \dots, \boldsymbol{y}_m\}$ of $m$ independent random vectors in $\mathbb{R}^d$. Let $|\boldsymbol{x}_i^T \boldsymbol{y}_a| \leq h = \mathcal{O}_m(1)$ for all $a$. Letting $Z_i$ be defined as in Equation (1), then for all $t > 0$*

$$\mathbb{P}\left(\left|\frac{Z_i}{m} - \mathbb{E}\left[\frac{Z_i}{m}\right]\right| \geq t\right) \leq 4e^{-(\sqrt{m}t/4eh)^2} .$$

Before proceeding with the proof of Theorem A.1, let us enunciate the following concentration theorem that we will use in our demonstration.

**Theorem A.2** ((Talagrand, 1995), (Ledoux, 2001), Corollary 4.10). *Given a random vector $\boldsymbol{\omega} \in [u, v]^n$ with independent entries and a 1-Lipschitz (for the euclidean norm) and convex mapping $g : \mathbb{R}^n \to \mathbb{R}$, one has the concentration inequality:*

$$\forall\, t > 0: \quad \mathbb{P}\left(|g(\boldsymbol{\omega}) - \mathbb{E}[g(\boldsymbol{\omega})]| \geq t\right) \leq 4e^{-t^2/4(u-v)^2} .$$

**Proof** (Theorem A.1). *Let $\boldsymbol{\omega}^{(i)} \in \mathbb{R}^m$ be a vector with entries $\{\boldsymbol{x}_i^T \boldsymbol{y}_a\}_{a=1,\dots,m}$. Then, the vector $\boldsymbol{\omega}^{(i)}$ satisfies the hypothesis of Theorem A.2 for all $i$ and for $v = -u = h$. We let $g$ be:*

$$g(\boldsymbol{w}^{(i)}) = \frac{1}{m} \sum_{a=1}^{m} e^{w_a^{(i)}} ,$$

then one can immediately verify that $Z_i = mg(\boldsymbol{\omega}^{(i)})$. We are left to prove that $g$ satisfies the hypotheses of Theorem A.2 as well. Firstly, $g$ is convex because it is the sum of convex functions. We now compute the Lipschitz parameter. Considering $\boldsymbol{\omega}, \boldsymbol{\omega}' \in [-h, h]^m$ one obtains the following bound:

$$|g(\boldsymbol{\omega}) - g(\boldsymbol{\omega}')| \overset{(a)}{\leq} \frac{1}{m} \sum_{a=1}^{m} \left| e^{\omega_a} - e^{\omega'_a} \right| \overset{(b)}{\leq} \frac{e}{m} \sum_{a=1}^{m} |\omega_a - \omega'_a| \overset{(c)}{=} \frac{e}{m} \cdot \mathbf{1}_m^T |\boldsymbol{\omega} - \boldsymbol{\omega}'|$$

$$\overset{(d)}{\leq} \frac{e}{m} \cdot \|\mathbf{1}_m\| \cdot \|\boldsymbol{\omega} - \boldsymbol{\omega}'\| = \frac{e}{\sqrt{m}} \|\boldsymbol{\omega} - \boldsymbol{\omega}'\| \ ,$$

where in $(a)$ we used the triangle inequality, in $(b)$ we exploited the fundamental theorem of calculus, in $(c)$ the $|\cdot|$ is meant entry-wise and finally in $(d)$ we used the Cauchy-Schwartz inequality. This set of inequalities implies that $\frac{\sqrt{m}g}{e}$ is 1-Lipschitz and is a suitable choice to apply Theorem A.2. Theorem A.1 is then easily obtained from a small play on $t$ and exploiting the relation between $Z_i$ and $g(\boldsymbol{\omega}^{(i)})$.

We remark that this result holds unchanged in the case $X = Y$. The proof can be easily adapted by singling out the negligible contribution given by $e^{\|\boldsymbol{x}_i\|^2} \leq e^h = \mathcal{O}_n(1)$, obtaining the same result.

**Theorem A.3.** *Consider a vector $\boldsymbol{x}_i \in \mathbb{R}^d$ and a set $\{\boldsymbol{y}_1, \ldots, \boldsymbol{y}_m\}$ of $m$ independent random vectors in $\mathbb{R}^d$. Let $|\boldsymbol{x}_i^T \boldsymbol{y}_a| \leq h = \mathcal{O}_m(1)$ for all $a$. Let $f_{i,m}$ denote the empirical distribution of $\boldsymbol{x}_i^T \boldsymbol{y}_a$ and assume $f_{i,m}$ converges in distribution to $f_i$, then,*

$$\lim_{m \to \infty} \mathbb{E}\left[\frac{Z_i}{m}\right] = \int_{-h}^{h} dt \ e^t f_i(t) \ .$$

Also in this case, before proceeding with the proof, let us enunciate the dominated convergence theorem that allows one to invert the integral and limit signs.

**Theorem A.4** ((Luxemburg, 1971)). *Let $F_1, \ldots, F_n$ be a sequence of Riemann-integrable functions – defined on a bounded and closed interval $[a, b]$ – which converges on $[a, b]$ to a Riemann-integrable function $F$. If there exists a constant $m > 0$ satisfying $|F_n(t) \leq M|$ for all $x \in [a, b]$ and for all $n$, then*

$$\lim_{n \to \infty} \int_{a}^{b} dt \ F_n(t) = \int_{a}^{b} dt \lim_{n \to \infty} F_n(t) = \int_{a}^{b} dt \ F(t) \ .$$

**Proof** (Theorem A.3). *The values $\boldsymbol{x}_i^T \boldsymbol{y}_a := \omega_a^{(i)}$ are a sequence of $m$ random variables with cumulative densities $F_{i,1}, \ldots, F_{i,m}$, where $F_{i,p}$ denotes the integral of $f_{i,p}$.*

$$\lim_{m \to \infty} \mathbb{E}\left[\frac{Z_i}{m}\right] = \lim_{m \to \infty} \frac{1}{m} \sum_{a=1}^{m} \mathbb{E}\left[e^{\omega_a^{(i)}}\right] = \lim_{m \to \infty} \mathbb{E}\left[e^{\omega_m^{(i)}}\right] = \lim_{m \to \infty} \int_{-h}^{h} dt \ e^t f_{i,m}(t)$$

$$\overset{(a)}{=} \lim_{m \to \infty} \left[ e^t F_{i,m}(t) \Big|_{-h}^{h} - \int_{-h}^{h} dt \ e^t F_{i,m}(t) \right] \xrightarrow{(b)}_{m \to \infty} e^t F_i(t) \Big|_{-h}^{h} - \int_{-h}^{h} dt \ e^t F_i(t) = \int_{-h}^{h} dt \ e^t f_i(t) \ .$$

In $(a)$ we performed an integration by parts; in $(b)$ we exploited the fact that convergence in distribution implies the pointwise convergence of the probability density function and we applied Theorem A.4 for $M = e^h$.

## B  Derivation of Equation (7)

We here report the derivation of Equation (7) from (6).

$$\mathcal{L} = -\sum_{i,j\in\mathcal{V}} P_{ij}\, \log\Big(\mathrm{SoftMax}(YY^T)_{ij}\Big) + \frac{1}{n}\sum_{i,j\in\mathcal{V}} \boldsymbol{y}_i^T\boldsymbol{y}_j$$

$$\stackrel{(a)}{=} -\sum_{i,j\in\mathcal{V}} P_{ij}(YY^T)_{ij} + \sum_{i,j\in\mathcal{V}} P_{ij}\, \log(Z_i) + \frac{1}{n}\sum_{i,j\in\mathcal{V}} (YY^T)_{ij}$$

$$\stackrel{(b)}{=} -\sum_{i,j\in\mathcal{V}} P_{ij}(YY^T)_{ij} + \sum_{i\in\mathcal{V}} \log(Z_i) + \frac{1}{n}\sum_{i,j\in\mathcal{V}} (\mathbf{1}_n\mathbf{1}_n^T)_{ij}(YY^T)_{ij}$$

$$\stackrel{(c)}{=} -\sum_{i\in\mathcal{V}} (PYY^T)_{ii} + \sum_{i\in\mathcal{V}} \log(Z_i) + \frac{1}{n}\sum_{i\in\mathcal{V}} (\mathbf{1}_n\mathbf{1}_n^TYY^T)_{ii}$$

$$\stackrel{(d)}{=} -\mathrm{tr}(PYY^T) + \sum_{i\in\mathcal{V}} \log(Z_i) + \frac{1}{n}\mathrm{tr}(\mathbf{1}_n\mathbf{1}_n^TYY^T)$$

$$\stackrel{(e)}{=} -\mathrm{tr}(Y^TPY) + \sum_{i\in\mathcal{V}} \log(Z_i) + \frac{1}{n}\mathrm{tr}(Y^T\mathbf{1}_n\mathbf{1}_n^TY)\ ,$$

where in $(a)$ we used the softmax definition and rewrote $\boldsymbol{y}_i^T\boldsymbol{y}_j = (YY^T)_{ij}$; in $(b)$ we used $\sum_{j\in\mathcal{V}} P_{ij} = 1$; in $(c)$ we used the fact that $YY^T$ is a symmetric matrix; in $(d)$ we leverage the trace definition; in $(e)$ we use the property of the trace $\mathrm{tr}(AB) = \mathrm{tr}(BA)$.

## C  Derivation of the gradient

We here derive the gradient expression as it appears in Equation (8). Note that in this derivation, the quantities $\boldsymbol{\mu}_\alpha$, $\Omega_\alpha$ are considered as constants, in a stochastic gradient descent fashion. We observed that this gradient form achieves better results in fewer epoch. Let us first rewrite the loss function of Equation (6). Following the passages detailed in Appendix B, we obtain

$$\mathcal{L} = -\sum_{i,j\in\mathcal{V}} \left(P_{ij} - \frac{1}{n}\right) \boldsymbol{x}_i^T\boldsymbol{x}_j + \sum_{i\in\mathcal{V}} \log(Z_i)\ .$$

We now introduce the approximation of Equation (4) and rewrite

$$\mathcal{L} \approx -\sum_{i,j\in\mathcal{V}} \left(P_{ij} - \frac{1}{n}\right) \boldsymbol{x}_i^T\boldsymbol{x}_j + \sum_{i\in\mathcal{V}} \log\ \sum_{\alpha=1}^{\kappa} n\pi_\alpha\ \exp\left\{\boldsymbol{x}_i^T\boldsymbol{\mu}_\alpha + \frac{1}{2}\boldsymbol{x}_i^T\Omega_\alpha\boldsymbol{x}_i\right\}$$

$$= -\sum_{i,j\in\mathcal{V}}\sum_{q=1}^{d} \left(P_{ij} - \frac{1}{n}\right) x_{iq}x_{jq} + \sum_{i\in\mathcal{V}} \log\ \sum_{\alpha=1}^{\kappa} n\pi_\alpha\ \exp\left\{\sum_{q=1}^{d} x_{iq}\mu_{\alpha,q} + \frac{1}{2}\sum_{q,p=1}^{d} x_{iq}\Omega_{\alpha,qp}x_{ip}\right\}\ .$$

We now compute the derivative with respect to $x_{kr}$ to obtain the respective gradient term.

$$\partial_{x_{kr}}\mathcal{L} \approx -\sum_{i,j\in\mathcal{V}}\sum_{q=1}^{d} \left(P_{ij} - \frac{1}{n}\right) \delta_{qr}\,[\delta_{ik}x_{jq} + \delta_{jk}x_{iq}]$$

$$+ \sum_{i\in\mathcal{V}} \frac{\sum_{\alpha=1}^{\kappa} \pi_\alpha\ e^{\boldsymbol{x}_i^T\boldsymbol{\mu}_\alpha + \frac{1}{2}\boldsymbol{x}_i^T\Omega_\alpha\boldsymbol{x}_i} \left(\sum_{q=1}^{d} \delta_{ik}\delta_{qr}\mu_{\alpha,q} + \sum_{q,p=1}^{d} \Omega_{\alpha,qp}\,[\delta_{ik}\delta_{qr}x_{ip} + \delta_{ik}\delta_{pr}x_{iq}]\right)}{\sum_{\alpha=1}^{\kappa} \pi_\alpha\ e^{\boldsymbol{x}_i^T\boldsymbol{\mu}_\alpha + \frac{1}{2}\boldsymbol{x}_i^T\Omega_\alpha\boldsymbol{x}_i}}\ .$$

We now use the notation $\mathcal{Z}_{i\alpha} = \pi_\alpha \; e^{\boldsymbol{x}_i^T \boldsymbol{\mu}_\alpha + \frac{1}{2}\boldsymbol{x}_i^T \Omega_\alpha \boldsymbol{x}_i}$ and compute the sums.

$$\partial_{x_{kr}}\mathcal{L} \approx -\sum_{i\in\mathcal{V}}\left(P_{ik} - \frac{1}{n}\right)x_{ir} - \sum_{j\in\mathcal{V}}\left(P_{kj} - \frac{1}{n}\right)x_{jr}$$

$$+ \frac{1}{\sum_{\alpha=1}^{\kappa}\mathcal{Z}_{k\alpha}} \cdot \sum_{\alpha=1}^{\kappa}\mathcal{Z}_{k\alpha}\left(\mu_{\alpha,r} + \frac{1}{2}\sum_{q=1}^{d}x_{kq}\Omega_{\alpha,rq} + \frac{1}{2}\sum_{q=1}^{d}x_{kq}\Omega_{\alpha,qr}\right)$$

Now we recall that $\sum_{\alpha=1}^{\kappa}\mathcal{Z}_{i\alpha} = Z_i/n$ and $M_{\alpha,r} = \mu_{\alpha,r}$ and that $\Omega_\alpha = \Omega_\alpha^T$.

$$\partial_{x_{kr}}\mathcal{L} \approx -\left[\left(P^T - \frac{1}{n}\mathbf{1}_n\mathbf{1}_n^T\right)X\right]_{kr} - \left[\left(P - \frac{1}{n}\mathbf{1}_n\mathbf{1}_n^T\right)X\right]_{kr} + \frac{1}{(\mathcal{Z}\mathbf{1}_\kappa)_k} \cdot \left([\mathcal{Z}M]_{kr} + \sum_{\alpha=1}^{\kappa}\mathcal{Z}_{k\alpha}[X\Omega_\alpha]_{kr}\right) \; ,$$

thus obtaining Equation (8).

## D  Experiment implementation details

We here report some implementation details in the experiments we conducted. This section complements the information in the main text when this is insufficient to reproduce our results.

### D.1  Node embeddings

In the experiments we test the node embedding problem for the task of community detection. To do so, we work with synthetic graphs generated from the degree corrected stochastic block model (DCSBM) (Karrer & Newman, 2011) that we here define.

**Definition D.1** (DCSBM). *Let $\omega : \mathcal{V} \to \{1, \dots, q\}$ be a class labeling function, where $q$ is the number of classes. Let $\mathbb{P}(\omega_i = a) = q^{-1}$ and consider two positive integers satisfying $c_{\text{in}} > c_{\text{out}} \geq 0$. Further Let $\theta \sim p_\theta$ be a random variable that encodes the intrinsic node connectivity, with $\mathbb{E}[\theta] = 1$ and finite variance. For all $i \in \mathcal{V}, \theta_i$ is drawn independently at random from $p_\theta$. The entries of the graph adjacency matrix are generated independently (up to symmetry) at random with probability*

$$\mathbb{P}(A_{ij} = 1) = \frac{\theta_i\theta_j}{n} \cdot \begin{cases} c_{\text{in}} & \text{if } \omega(i) = \omega(j) \\ c_{\text{out}} & \text{else} \end{cases}$$

In words, nodes in the same community ($\omega(i) = \omega(j)$) are connected with a higher probability than nodes in different communities. From a straightforward calculation, the expected degree is $\mathbb{E}[d_i] \propto \theta_i$, thus allowing one to model the broad degree distributions typically observed in real networks (Barabási & Albert, 1999). Given this model, the community detection task consists in inferring the node label assignment from a realization of $A$. It was shown that this is theoretically feasible (in the large $n$ regime) if and only if $\alpha = (c - c_{\text{out}})\sqrt{\frac{\mathbb{E}[\theta^2]}{c}} > 1$. This is the $\alpha$ parameters appearing in the main text. In the simulations the $\theta_i$'s are obtained by: i) drawing a random variable from a uniform distribution between 3 and 12; ii) raising it to the power 6; iii) normalize it so that $\mathbb{E}[\theta] = 1$. This leads to a rather broad degree distribution, even if it maintains a finite support.

For all three methods under comparison we obtain the embedding vectors from $A$ and then cluster the nodes into communities by applying *k-means* and supposing that the number of communities $q$ is known. The algorithm of (Dall'Amico et al., 2019; 2021) obtains the embedding by extracting a sequence of eigenvectors from a sequence of parameterized matrices that are automatically learned from the graph and does not require any parametrization. The `DeepWalk` and `EDRep` algorithms generate embeddings in $d = 32$ dimensions. We observed that the results are essentially invariant in a large spectrum of $d$ values for both embedding algorithms.

### D.2 Dynamically aware node embeddings

This experiment features three main steps: i) the creation of the supra-adjacency matrix from a temporal network; ii) the creation of an embedding based on this matrix; iii) the reconstruction of a dynamical process taking place on the network. We now describe these steps in detail.

**Definition of the supra-adjacency matrix**

We consider a temporal graph collected by the SocioPatterns collaboration. This dataset describes face-to-face proximity encounters between people at a conference. The data are recorded with a temporal resolution of 20 seconds and correspond to interactions within a distance of approximately 1.5 meters.[7] The dataset contains approximately 3000 different time-stamps. Following the procedure of (Sato et al., 2021) we aggregate them into $T = 180$ windows of approximately 15 minutes each. We thus obtain a temporal graph that is represented as a sequence of weighted temporal edges $(i, j, t, w_{ijt})$, indicating that $i, j$ interacting at snapshot $t$ for a cumulative time equal to $w_{ijt}$. We say that a node is active at time $t$ if it has neighbors at time $t$. We let $t_{i,a}$ be the time at which node $i$ is active for the $a$-th time. Given this graph, we then build the supra-adjacency matrix that is defined as follows.

**Definition D.2** (Supra-adjacency matrix). *Consider a temporal graph represented as a sequence of weighted temporal edges $(i, j, t, \omega_{ijt})$. We define a set of "temporal nodes" given by all the pairs $i, t_{i,a}$ where $i$ is a node of the temporal graph and $t_{i,a}$ is the time of the $a$-th appearance of $i$ in the network. We denote this set $\mathcal{D}$, formally defined as*

$$\mathcal{D} = \{(i, t_{i,a}) \ : \ i \in \mathcal{V}, \exists \ j \in \mathcal{V} \ : \ (i, j, t_{i,a}) \in \mathcal{E}\},$$

*where $\mathcal{V}$ is the set of nodes and $\mathcal{E}$ of temporal edges. The cardinality of this set is $D = |\mathcal{D}|$. We then define a directed graph with $\mathcal{D}$ being the set of nodes. The directed edges are placed between*

$$\begin{cases} (i, t_{i,a}) \to (i, t_{i,a+1}) & \text{self connections} \\ (i, t_{i,a}) \to (i, t_{j,b+1}) & \text{if } t_{i,a} = t_{j,b} \text{ and } (i, j, t) \in \mathcal{E} \\ (i, t_{j,b}) \to (i, t_{a,a+1}) & \text{if } t_{i,a} = t_{j,b} \text{ and } (i, j, t) \in \mathcal{E}. \end{cases}$$

*The supra-adjacency matrix $A_{\text{supra}} \in \mathbb{R}^{D \times D}$ is the adjacency matrix of the graph we just defined.*

**Creation of the embedding**

Given $A_{\text{supra}}$, the authors generate an embedding with the `DeepWalk` algorithm. Here, the random walker can only follow time-respecting paths, by construction of the matrix. We let the random walks have length equal to 15 steps and deploy also in this case the embedding dimension $d = 30$. For `EDRep` we use the row-normalized version of $A_{\text{supra}}$ as $P$. As a result we obtain an embedding vector for each $(i, t_{i,a})$ pair.

**Reconstruction of a partially observed dynamic process**

Following (Sato et al., 2021) we consider an epidemic process taking place on the temporal network. We use the SIR model (Keeling & Rohani, 2011) in which infected nodes (I) can make susceptible nodes (S) to transition to the infected state with a probability $\beta$ if they are in contact. Infected individuals then recover (R) with a probability $\mu$ and are unable to infect or get infected. We run the SIR model letting all nodes to be in the S state at the beginning of the simulation and having one infected node. The experiment is run with $\beta = 0.15$ and $\mu = 0.01$ and it outputs the state of each node at all times, i.e., for all $(i, t_{i,a})$.

We then suppose to observe a fraction of these states (every node is expected to only be observe once) and obtain a binary variable for each $(i, t_{i,a})$ indicating whether node $i$ was infected at time $t_{i,a}$. Exploiting the embeddings, we train a logistic regression model to predict the state of the node in each unobserved time and compare the predicted number of infected individuals against the observed ones. We repeat the experiment for 50 different realizations of the observed training set.

---

[7]The experiment is described in (Cattuto et al., 2010). The data are shared under the Creative Commons Public Domain Dedication license and can be downloaded at http://www.sociopatterns.org/datasets/sfhh-conference-data-set/.

