# OpenReview forum: "Learning distributed representations with efficient SoftMax normalization"
_TMLR — Accepted by TMLR_

### Review · Reviewer_2r1x · 2024-11-10

**Summary Of Contributions:**

The paper presents a method to efficiently compute the softmax attention scores in linear time. Particularly, it aims to efficiently compute the normalization constant for these attention scores, which is computationally time-consuming. The work relies on concentration bounds involving the normalization constant, assuming that the corresponding vectors have bounded norms. The paper showed that it is possible to improve efficiency by using a mixture of Gaussian parameterization and estimating the corresponding Gaussian parameters. Empirically they show that this formulation has a better approximation quality than existing baselines like Performers. Based on this observation, the paper presents an algorithm to learn feature representations given similarity values in the corresponding space.

**Audience:**

Yes

**Claims And Evidence:**

No

**Requested Changes:**

1. Please respond to the questions in the weakness section.
2. The paper in its current state would require some rewriting to make it more presentable to the reader.
    * Specifically, the paper should clearly state the problem that it is trying to solve up front in the abstract and introduction, whether it is an efficient attention mechanism or estimating feature vectors given a kernel space.
    * The mathematical notations should be simplified to help the readers. For example, several notations are used when defining the probabilities $P$: $\mathbf{p}_0$ $\mathbf{p}_j^{(i)}$, $\mathbf{p}^{(i)}$, $\mathbf{p}_{0, j}$. These notations are difficult to follow and should be simplified.
    * There should be more justifications for key assumptions in the paper. For example, it is unclear to me why $x^T_i y_a$ would converge to a value $\omega_i$ that is independent of $a$. The paper should also refer to the appendix for the proof of a corresponding mathematical result.
    * The experimental section also needs to be improved by incorporating detailed annotations for the figures.

**Strengths And Weaknesses:**

**Strengths**:

1. The paper tackles an important problem of efficiently computing attention scores in linear time.
2. The paper presents experiments in different settings with relevant baselines.


**Weaknesses**:
1. The paper is quite hard to follow especially because of the math notations used in it. Moreover, the motivation seems to change from the introduction. From the abstract, the reader expects an efficient algorithm for attention computation, but the main paper presents an optimization algorithm to extract the embeddings given an attention matrix.
2. Several of the mathematical notations and assumptions need more clarification. See below:
    * Is the convergence assumption in Theorem 2.2 realistic? Why should $x_i^Ty_a$ converge to the same value $\omega_i$ for all $y_a$?
    * $f_i$ is not defined in Theorem 2.2.
    * The paper should have pointers to the corresponding proofs in the appendix.
    * The proof of Theorem 2.2 is hard to follow. It starts off with $\omega_{a}^{(i)}$ instead of $\omega_i$. Moreover, $f_i$ or $F_{i, m}$ were never defined.
    * The approximation in the unnumbered equation after Theorem 2.2 can vary quite a bit depending on the value of $h$. There should be some assumption about $h$ or error analysis of this approximation as a function of $h$.

3. The main problem formulation in Section 3.1 is quite difficult to follow. See below:
    * why do we care about a probability distribution over the embedding space? The paper so far discussed efficient computation of the attention normalization constant.
    * The primary optimization objective is shown in Equation 4. What does $p_{0, j}$ indicate? Because this term is undefined, I did not understand the regularization term and its utility.

4. Section 3.3. The description of the algorithm is unclear. Step 1, which vectors do we cluster using the k-means algorithm?

5. The text refers to Figures 3A,B,C, and D but they are not defined in the diagram.

6. Several pieces of text in the paper are unclear. See the comments section.

7. Given that the main contribution of the paper is improving efficiency, there is little empirical support for the time improvement. Figure 2, only compares with the brute force approach. The legends are missing from the time results in Figure 3.






**Comments**:

“Note that this approximation does not require the embedding vectors to be, themselves, drawn from a multivariate Gaussian distribution.” — the commas should be removed from this sentence.


“We aim at giving a distributed representation of {p(i)}i∈V , in the form of an embedding matrix X ∈ Rn×d preserving the relational patterns encoded by the probability distributions.” — I’m not sure what this sentence implies.

---

> ### Author Response · Authors · 2024-11-13
>
> We thank the reviewer for their comments and gladly observe that they mostly pertain to exposition issues and not to the core results. We detail a point-by-point answer to the *weaknesses*.
>
> 1.
> * [notation]. We apologize for the confusion created by the notation. We denoted in bold-face the vector $\mathbf{p}^{(i)}$ and its $j$-th element as $p^{(i)}\_j = P\_{ij}$ in standard font. We felt that this choice made the narration smoother, but we will fix this problem by only using the simpler matrix notation.
> * [motivation]. We think that the perceived mismatch between abstract, introduction and main is because of the unclear exposition of the embedding problem (see also point 3). We thus plan to fix this issue by better introducing the embedding strategy. Instead, we believe that the abstract and introduction well reflect our contribution:
>     * Abstract: "we propose a linear-time approximation of the attention score normalization constants for embedding vectors with bounded norms": this is the content of Sec. 2
>     * Abstract: "From this result, we design a linear-time and task-agnostic embedding algorithm based on the optimization of the attention scores": this is the content of Sec. 3
>
>     Moreover, we show how to efficiently compute the attention normalization - and not attention itself - by using the logarithm of the score attention matrix as explained in the introduction. By improving the explanation of Equation 4, the abstract and introduction should feel more appropriate to describe our contribution.
>
> 2.
> * [convergence in Theorem 2.2]. We apologize for the lack of clarity on this point. We consider $\{\mathbf{y}\_1,\dots,\mathbf{y}\_m\}$ to be a set of random vectors drawn from an *unknown* distribution. For a fixed $\mathbf{x}\_i$ we look at $\{\mathbf{x}\_i^T\mathbf{y}\_1,\dots\mathbf{x}\_i^T\mathbf{y}\_m\}$ and denote with $f\_{i,m}$ their empirical distribution (this notation only appears in the Theorem's proof). We finally assume the loosest type of convergence of $f_{i,m}$ to $f_i$, the limiting *unknown* distribution of the scalar product. We plan to:
>     * Better explain the framing under which the embeddings are considered as random vectors.
>     * Drop the notation $\omega_i$ which is misleading and focus on the empirical distribution of the scalar product.
>
>     As explained in the Theorem's proof, $F$ is the integral of $f$ and, likewise $F_{i,m}$ the integral of $f_{i,m}$. This will be further clarified.
> * [pointers to the proofs]. Right before Sec 2.1 we wrote "The proofs of Theorem 2.1, 2.2 are given in Appendix A." We may add pointers near the theorems' ununciation.
> * [role of h]. The goodness of the approximation depends on $h$ as per the concentration result of Th 2.1, where we formulate the assumption on $h = O_m(1)$. We will put more emphasis on this important point.
>
> 3.
> * [embedding problem]. Thank you for pointing out the lack of clarity on this point. Our objective is not to create a probability distribution over the embedding space. The values $P_{ij}$ are given as inputs (see Algorithm 1), and the loss function is optimized over $Y$. $P$ describes the affinity patterns between item pairs that are learned by the embedding vectors. We plan to better motivate this loss function, relating it to the vast literature that uses similar approaches.
> * [role of $\mathbf{p}_0$]. Its scope is enforcing embeddings with a null average, as explained after Equation 4. Since $P$ is non-negative, we observed that without this term the embedding vectors tend to align to non-negative values as well, hampering the embedding quality. Moreover, we realized that we only explored the choice $\mathbf{p}_0 = \mathbf{1}_n/n$ and we plan to drop this notation and use $\mathbf{1}_n/n$ instead.
>
> 4. *k-means* algorithm is applied to the embedding matrix $X$ that appears in ${\rm SoftMax}(XX^T)$.
>
> 5. We apologize for the inconvenience and we will add the labels to the figure. The letters are A, C, E (first row), B, D, F (second row).
>
> 6. We will riformulated this sentence more clearly.
>
> 7. The legend in the bottom left panel of Fig. 3 is the same as the top left panel and it shows that Node2Vec is almost 5 times slower than EDRep. Concerning efficiency, there are two comments:
>     * We believe our main contribution goes beyond efficiency. There is a vast literature (as hinted in point 3) that intends or claims to use loss functions as in Eq. 4 but does not in practice because of its complexity. Our contribution is showing the possibility of efficiently optimizing these cost functions.
>     * As a by-product, our implementation is faster than the baselines which are linear-time algorithms with *different* loss functions. Note that we compared with the most efficient implementations available, typically coded in C++, unlike our Python code. For instance, the implementation we used for Node2Vec is by far faster than the official Python release.

---

### Review · Reviewer_iPsW · 2024-12-23

**Summary Of Contributions:**

Paper studies \*2vec-style embedding models like word2vec or node2vec. Although title has the term "attention" in it, this paper exclusively studies \*2vec-style embedding models where pairwise interaction probabilities between two entities i and j are modelled as $q_{j}^{(i)} = \exp(x_i^\top x_j)/\sum_{k \in \mathcal{V}} \exp(x_i^\top x_k)$ where $\mathcal{V}_i$ is the set of $n$ entities under consideration and $x_i \in \mathbb{R}^d$ is the embedding of entity $i$. Note that authors do not particularly consider or run experiments on attention mechanisms in Deep Learning models which makes the title a bit misleading.

Computing $q_{j}^{(i)}$ for all $i \in \{1, \ldots, n\}$ even for a fixed $j$ requires $O(n^2 d)$ computations as the denominator requires the computation of $n^2$ dot products. Authors propose to sidestep this computation by modeling distribution of current embedding vectors as a mixture of $\kappa$ Gaussians. This reduces the complexity of computing the denominator to $O(n \kappa d)$ for a fixed $j$.

Next authors use this strategy to provide an algorithm EDRep for solving \*2vec-style embedding problems more efficiently than naive implementations. Authors provide a set of comprehensive experiments to study the effectiveness of their mixture of Gaussian approximation and the EDRep algorithm on three \*2vec-style embedding problems.

**Audience:**

Yes

**Claims And Evidence:**

No

**Requested Changes:**

Please address the concerns mentioned above.

**Strengths And Weaknesses:**

### Strengths:
1. Paper provides some theory to justify their algorithm, although it is quite standard. The main idea is showing that the normalization constant in the denominator above converges fast to its mean which can be explicitly computed if we assumed embeddings come from a parametric mixture of Gaussians.

2. Given experimental results are very rigorous. It measures a variety of metrics, computational costs and provide error margins

3. Paper is well written and decently motivated. However, it missed the contemporary context (see below).

4. Experimental results are very promising and show that the proposed approximation mechanism works well for the problems studied.

5. EDRep algorithm provided seems novel (although straight forward) and it is effective in reducing computational cost

### Weaknesses:
1. Theory seems to be a missing both minor and major details. What is $h$ in Thm 2.2? How is (2) derived? How is (3) derived and why is K-means clustering the solution?

2. Paper assumes that $\mathcal{V}_i$ consist of embeddings which are independently chosen. This works when $\mathcal{V}_i$ is large enough or if it is the whole universe of entities (like full vocab of tokens). However, when $\mathcal{V}_i$ is restricted to small window of entities (e.g. context of words around a word in a sequence) independence may not hold true, hence the approximation technique might fail. Similarly, any kind of position encoding used in transformers would break the independence assumption.

3. Many of the experiments (Fig 1 and 2) provide absolute error but fail to provide relative error which is more important for such calculations.

4. Experiments only study quite old \*2vec style methods and they do not study actual attention mechanism in deep learning models. Title is also misleading in that sense. It might also be true that \*2vec style methods are not that powerful when compared to transformer/GNN style models.

5. If the empirical entity interaction probability matrix $P \in [0,1]^{n \times n}$ is not sparse, then during training we would require the computation of $O(n^2)$ $q_j^{(i)}$'s which requires $O(n^2d)$ computations and the proposed attention mechanism won't produce linear complexity. In that case it is not clear how $P$ can be decomposed into product of $m$ sparse matrices to reduce the complexity.

6. Paper do not discuss or compare with many works on reducing attention complexity [1, 2]. See [3] for comprehensive discussion. Therefore, it is not clear where the proposed method sits in the current literature space.

7. It is not clear whether *2vec style baselines in Sec 4 use any computationally efficient techniques. For example, in standard word/node-2vec only a small number/subset of positive and negative examples are chosen per sample which effectively guards the complexity from blowing up quadratically. So it isn't clear how such simpler techniques compare against the proposed method.

[1] Zaheer, M., Guruganesh, G., Dubey, K. A., Ainslie, J., Alberti, C., Ontanon, S., ... & Ahmed, A. (2020). Big bird: Transformers for longer sequences. Advances in neural information processing systems, 33, 17283-17297.

[2] Dao, T., Fu, D., Ermon, S., Rudra, A., & Ré, C. (2022). Flashattention: Fast and memory-efficient exact attention with io-awareness. Advances in Neural Information Processing Systems, 35, 16344-16359.

[3] Wan, Z., Wang, X., Liu, C., Alam, S., Zheng, Y., Liu, J., ... & Zhang, M. (2023). Efficient large language models: A survey. arXiv preprint arXiv:2312.03863.

---

> ### Author Response · Authors · 2025-01-03
>
> We thank the reviewer for their work and the appreciation of our contribution. Following their suggestion, we will implement some modifications to the paper that will be shared after the last review is received.
>
> * 1a) The value $h$ is a constant (appearing also in Th 1, which is referenced by Th 2), independent of $n$. We will recall this notation to make the enunciation self-contained.
>
>   1b) Eq (2) is obtained by solving the Gaussian integral in the previous (un-numbered) equation. We will add a reference to make this point clearer.
>
>   1c) Eq (3) denotes the relative size, mean vector, and covariance matrix of all embedding vectors within a class. We plan to better discuss the rationale of our approach in a new version of the paper. Briefly, we approximate the distribution of the scalar product $f_i$ with a mixture of $\kappa$ Gaussians: this approximation leads to an arbitrarily small error, provided $\kappa$ is large enough. However, we need a simple way to obtain good results with small $\kappa$. Entities can often be grouped according to semantic or functional similarities and are represented with similar embedding vectors. The intuition is to identify these groups - inferred with k-means - with indices in the Gaussian mixture exploiting the higher similarity among embeddings within the group. This reasoning is kept at a heuristic level (as declared in the beginning of Sec 2), but the empirical analysis shows its efficacy.
>
> * 2 Assuming $\mathcal{V}$ is large is necessary for our approximation. It follows from Eq (1) and $m = O_n(n)$. However, this is also the regime of interest in which computing $Z_i$ is expensive. We plan to put more emphasis on this point.
>
> * 3 We updated Figure 1 using absolute instead of relative error. The results are very similar and, in some occasions (such as the 4th-row, rightmost column), better than the relative error. On the other hand, Figure 2 shows the absolute error: it is only rescaled to make it independent of $n$.
>
> * 4, 6, and notes in the summary. Our main contribution is to provide a linear-time approximation of the attention score **normalization** and not of attention itself. We are sorry the reviewer finds the title misleading, but we feel it fairly describes our contribution. The expression "attention scores" is commonly adopted in the literature (e.g. https://openreview.net/pdf?id=R8sQPpGCv0) to denote $Softmax(XY^T)$.
>  From the introduction, we frame our contribution in the realm of 2Vec-type algorithms and only mention transformers as an example of algorithms that optimize the attention score matrix. We underline that our results do not allow one to directly obtain an efficient transformer because we exclusively show how to efficiently compute the score normalization. As we explain in the text, this is the computational bottleneck in 2Vec algorithms like the one we consider and a comparison with efficient transformer-based approaches would be inappropriate, because we are considering different embedding problems. We plan to add a discussion in the conclusion of our paper on this aspect as a limitation of our approach. Nonetheless, in Section 2.2 we compare our approximation with two popular efficient transformer methods -- also referenced in [3] -- that, as we do and unlike other methods like Big Bird, approximate rather than modify the loss function. Figure 1 shows our approximation largely outperforms the existing ones. Importantly, we are not claiming that the other approaches are worse than approximating Eq (4) and determining the best approach is likely task-dependent. On the opposite, we show how to efficiently optimize the loss function in Eq (4). Part of this argument appears in the last paragraph of the Introduction, but we plan to improve it to better clarify our relation with the literature and add further references to the literature on efficient transformers.
>
> * 5 Indeed, the embedding algorithm is linear in $E$ which may grows super-linearly with $n$. However, the title refers to a "linear-time attention score normalization": the normalization approximation of Eq (2) is independent of $P$ and its complexity grows linearly with $n$. In other words, the complexity of EDRep depends on $P$ (an input that we cannot change) and on the attention scores (which we bring down to $O(n)$). In practice, to keep the algorithm efficient, $P$ can be decomposed into the product of sparse matrices by design (as we do in section 4.1) or, following (Le Magoarou & Gribonval, 2016), one can write $P$ as the product of sparse matrices $P\approx \prod_i P_i$. In this case, the complexity runs with $O(\sum_{i=1} E_i)$ instead of $O(n^2)$, where $E_i$ is the number of non-zero entries of $P_i$.
>
> * 7 The comparison is performed against the computationally efficient implementation of 2Vec algorithms whose complexity runs linearly with $n$. Note that we compared with the most efficient implementations available, typically coded in C++, unlike our Python code.

---

### Review · Reviewer_JiP6 · 2025-02-11

**Summary Of Contributions:**

This paper studies approximations of the "attention score matrix", $\textrm{SoftMax}(XY^T)$.

The paper proposes:
1) a mixture of gaussian-based, linear in the number of rows of $Y$ approximation to the partition function used for normalizing the rows of $XY^T$ in the softmax function.
2) An algorithm EDRep for learning embedding in the "2Vec" style of methods (e.g., matrix factorization).
3) Performs analysis of the proposed approach and empirically investigates EDRep in diverse applications.

**Audience:**

Yes

**Claims And Evidence:**

No

**Requested Changes:**

Please comment on / address W1-W3 and address attention score matrix terminology.

**Strengths And Weaknesses:**

Strengths:
* Clearly written overall, easy to follow paper
* Nice presentation of a simple method with diverse applications
* Interesting clustering-based approximation to softmax

Major concerns:
* W1: Presentation of Attention Score Matrix - The baseline methods compared in section 2 ([Random Feature Attention](https://arxiv.org/abs/2103.02143) and [Performers](https://arxiv.org/pdf/2009.14794) are used to approximate the attention score matrix in a transformer's attention layer. In my opinion, this is a significantly different setting than the one proposed in this paper, with different computational tradeoffs. I have the following concerns surrounding this:
  * W1.1: Why are these the right baseline methods to consider for the setting used in this paper? What about considering more classic density estimation methods? What about simple sampling or top-k based approximations? Or simple matrix factorizations style approximations?
  * W1.2: Another concern about these as baselines, is that they do not share the same $m\rightarrow\infty$ assumption in many use cases.
  * W1.3:  In measuring the computational cost of the mixture-of-gaussian based method for softmax approximation (E.g. Equation 2), the $O(d^2)$  ($d$ as in dimension of the vectors), is discussed briefly, but then quickly dropped. This is a quite significant term. It is especially significant for the use case of approximating the attention matrix in a transformer layer. This seems to make the method well suited for the $d << m$ case, but less well suited for the $d < m$ case. Should we really not be so concerned about this quadratic term? I guess the thought is $d$ might be on the order of a few hundred so it is ok? Nevertheless, when comparing to methods without this term, it seems significant to mention.
* W2: Approximation quality guarantees of EDRep
 * While the derivation of the EDRep algorithm is nice, it feels largely incomplete because it is missing, as I understand statements about either: generalization, quality of approximation / representational capacity, or convergence.
  * It is not entirely clear to me the advantages of EDRep over alternatives, alternative models for the data etc. Especially for many of the applications that are considered in this paper, e.g. the "2Vec" use case (where countless alternative parameterizations have been considered).
* W3: Missing related work:
* While the paper does cover a good deal of related work, I think there are significant gaps for instance:
  * Baharav et al, 2024 [https://proceedings.neurips.cc/paper_files/paper/2024/file/d52dbd66219dc4e432e0bd4f9c25c4c3-Paper-Conference.pdf]
  * Tree-based methods such as [Blanc et al 2018](https://proceedings.mlr.press/v80/blanc18a/blanc18a.pdf)
  * For Word2Vec relationship specifically: https://aclanthology.org/Q15-1016.pdf

More minor concerns:
* The complexity of the proof is more limited than I would expect for TMLR, yet the paper reads in a largely theoretical way. I think a better presentation could be to strong emphasis the empirical contributions (once the above concerns are taken care of) and then using the theoretical analysis to support the empirically motivated work.
* The proposed method is quite simple, but if presented in full level of rigor amidst the kinds of baselines and open questions described in above weaknesses, might be enough for a full paper contribution.
* The choice of clustering algorithm seems important for the approximation quality, especially theoretically. Why is K-means the right choice given that the full covariance matrix is used? Please provide more explanation and details
* It would be nice to give a bit of intuition about Equation 2, given this paper is generally accessible, i think this would be important to readers. E.g. covariance operator?
* Some of the writing in the introduction, could be made more crisp: How common is the terminology: attention score matrix SoftMax(XY T ) ? I have seen it for an attention layer, but perhaps when defining it broadly like this, good to put citations? Also, I found the sentence " The optimization of the attention scores motivates several embedding algorithms such as Word2Vec Mikolov et al. (2013) and the many “2Vec”-type algorithms that followed". Hard to parse and a bit vague. It is not obvious at this point what you mean by optimization of the attention scores. The use of “several” and “many” is confusing also.

---

> ### Author Response · Authors · 2025-02-24
>
> We thank the reviewer for the careful and constructive feedback and the overall appreciation of our work.
>
> ## Major concerns
>
> * W1. The selected methods were chosen because they could be fairly compared in our setting and without any modification. As we made clearer in the new version of the article, while several methods exist to approximate softmax, not all of them consider the optimization problem from the same angle as we do. Commonly, efficiency is part of an optimization task, and looking at the performance in terms of estimation accuracy can be either unfair or directly impossible. The selected low-rank approximation methods, instead, allow a simple method to directly estimate the normalization constants (even if that is not their primary objective) as we do. Our approach is very "vertical" on the problem we considered, which makes it efficient but not directly adaptable to transformer-based architectures as we more clearly state in the discussion.
>
> * W1_bis. We thank you for pointing out some confusion made in interpreting the role of the parameters $m, n, d$. As we now explain more in detail, the condition $m = O(n)$ is relevant because it entails the hardest regime in which computing the normalization constants scales quadratically with the problem size. On the other hand, we consider $d \ll n$, which, in our case, is necessary to hope for a linear-time algorithm. This explains why the $O(d^2)$ complexity is not harmful, but we added some comments on it. Moreover, we also commented on (Baharav et al, 2024) which considered a different angle from ours. They provide an estimation of the normalization constant with a complexity scaling sub-linearly in $d$ but super-linearly in $n$. Their algorithm, hence, does not scale well in our regime (and vice versa).
>
> * W2. In this manuscript, we choose to privilege an approach with very few assumptions, at the cost of not providing an error bound. Instead of proceeding with hard-to-verify assumptions, we validated our results empirically, and "rigorously" as noted by reviewer iPsW. We made this point clear in the first version among the limitations of our work and we insisted even more on evidencing that part of our results are data-driven. The reviewer, however, refers to an analysis of the EDRep algorithm. Given our approximation, this algorithm simply runs gradient descent (we compute the gradient analytically) and does not perform any other operation. The loss function Eq (4) is simple and our whole point is to show that we can optimize it efficiently, not that we are proposing one more 2Vec algorithm. Moreover, there already exist works such as (Jaffe et al, 2020) in which the authors studied the 2Vec loss function, while actually analyzing Eq. 6 in our paper.
>
> * W3. We added and discussed the references indicated by the reviewer and other relevant works that helped us frame our proposed method better.
>
> ## Minor concerns
>
> * We have moved the theorems to the appendix and rewrote the main contribution of the article in a less formal dress, insisting more on the intuitions, and empirical validations.
>
> * We reformulated the derivation of the multivariate Gaussian approximation and better motivated the use of k-means, by showing that the k-means objective is an upper bound to a "total variance" quantity. We argue that a lower within-class variance leads to better results, hence the reason why using k-means.
>
> * We rewrote parts of the text (especially in the first two sections) to enhance the readability. Moreover, accounting for the comments raised by the other reviewers we chose to change the attention nomenclature (and the title) and use "softmax scores" instead.

---

> > ### Comment · Reviewer_JiP6 · 2025-04-15
> >
> > Thank you authors for your thorough revisions and attention to my comments and feedback. This is greatly appreciated. I apologize for my delay in replying to you.
> >
> > To me, the paper seems much more technically sound and compelling in its current form than its previous revisions.
> >
> > I still feel a bit unsatisfied and have some lingering doubts about the evaluation in Figure 1 and the choice of methods compared there. I think adding further evaluations of methods that are more focused on softmax function approximation and less on efficient transformers. However, perhaps given the tone of the new paper draft though, it is okay. And further the results in Figure 3 are supportive and thorough enough for the claims to be considered upheld.

---

### Author Response · Authors · 2025-02-24

We have uploaded a new version of the paper. In the submission form above we detailed the main changes that were performed.

---

### Decision · Action_Editor_L1A5 · 2025-04-28

**Recommendation:** Accept with minor revision

**Comment:**

The paper proposes a heuristic linear-time approximation method for efficiently computing normalization constants of softmax functions for embedding vectors with bounded norms.

Reviewers raised initial concerns regarding clarity, theoretical guarantees, and the selection of baseline comparisons. Specifically:
- confusion about whether the approach was designed for transformer architectures or more general embedding methods, recommending clearer framing and explicit statements of the scope.
- necessity for additional comparisons
- notational clarity

The authors addressed these concerns by clarifying their focus, explicitly distinguishing their contributions from transformer-specific methods, justifying their methodological choices, and improving overall manuscript readability. As a result reviewers considered the paper is thorough, is adequate in its descriptions and investigations of ideas, and is a sufficiently informative artifact of information to be shared with the TMLR community.

*Requested changes:*
That said, reviewers have remaining doubts about the choices of methods compared to in one of the principle results in the paper. In figure 1, the authors compare to RFM & Performers when approximating softmax normalization constants in an isolated (non end-to-end transformer) setting. However there is a body of other work on directly approximating the softmax that could have been compared (e.g., simple top-k baseline?, fitting hierarchical softmax or Blanc & Rendle (2018)? or more common matrix factorization / low-rank approaches?). The concern is that comparing with methods mainly intended for use in one way, evaluated for another way, can lead to confusion. The methods compared are designed for efficient attention computation in transformers. It is my understanding that effectively estimating the softmax normalization is not a necessary condition for these approaches, given that they are used in end-to-end systems. So please add a comparison with respect to softmax approximation literature.

**Audience:**

Yes, embedding learning community and perhaps efficient transformer community.

**Claims And Evidence:**

In the initial submission, there was a confusion around claim of efficient transformer design with approximating the softmax while performing experiments on "2Vec" style approaches. The new version claims are supported by provided experiments and theory.

---

> ### Author Response · Authors · 2025-05-28
>
> Dear Editor,
>
> we thank you for your work and the assessment of our paper. We updated the camera-ready version by adding the requested comparison with more standard softmax approximations. In particular, we considered sampling (Blanc & Rendle, 2018), low-rank approximations (Drineas et al., 2005), and top-k approximations (Mussmann et al., 2017). We fixed the free parameters of these models to obtain execution times comparable with our method. The accuracies obtained by our approximations are higher than those obtained by the competing methods. Only in one dataset, the low-rank approach is more accurate than our proposed approximation, but on other datasets, it is even worse than the other methods under comparison. The new results hence confirm and strengthen the claims in the prior version of the paper.
> We also realized that the confusion of the AE related to the comparison with (Peng et al., 2021) is because this is not the appropriate reference for the technique considered. In fact, (Peng et al., 2021) develop an efficient transformer and do not aim to simply obtain an efficient estimation of softmax. The technical tool used by the authors are random features, for which they cite (Rahimi & Recht, 2007) which is a more appropriate reference. On the other hand, (Choromanski et al., 2020) also develop an efficient transformer, but they also introduce an alternative method to create random features. We think random features-based estimators are also a relevant baseline, and we thus chose to keep them in the comparison, but we explained them and referenced them better.